# Stromal FOXF2 suppresses prostate cancer progression and metastasis by enhancing antitumor immunity

Deyong Jia [1], Zhicheng Zhou[1], Oh-Joon Kwon [1], Li Zhang[1], Xing Wei[1], Yiqun Zhang [2], Mingyang Yi[3], Martine P. Roudier[1], Mary C. Regier[4,5], Ruth Dumpit[6], Peter S. Nelson [6], Mark Headley[6], Lawrence True[5], Daniel W. Lin[1], Colm Morrissey[1], Chad J. Creighton[2] & Li Xin [1,4] ✉

Cancer-associated fibroblasts (CAFs) mediate an immunosuppressive effect, but the underlying mechanism remains incompletely defined. Here we show that increasing prostatic stromal *Foxf2* suppresses the growth and progression of both syngeneic and autochthonous mouse prostate cancer models in an immunocompetent context. Mechanistically, Foxf2 moderately attenuates the CAF phenotype and transcriptionally downregulates *Cxcl5*, which diminish the immunosuppressive myeloid cells and enhance T cell cytotoxicity. Increasing prostatic stromal *Foxf2* sensitizes prostate cancer to the immune checkpoint blockade therapies. Augmenting lung stromal *Foxf2* also mediates an immunosuppressive milieu and inhibits lung colonization of prostate cancer. *FOXF2* is expressed higher in the stroma of human transition zone (TZ) than peripheral zone (PZ) prostate. The stromal FOXF2 expression level in primary prostate cancers inversely correlates with the Gleason grade. Our study establishes Foxf2 as a stromal transcription factor modulating the tumor immune microenvironment and potentially explains why cancers are relatively rare and indolent in the TZ prostate.

The tumor-immune microenvironment (TIME) plays a crucial role in tumor initiation and progression[1]. A role of transcription factors in modulating TIME has been intensively studied. Common drivers of tumorigenesis can mediate tumor cell-intrinsic inflammatory signaling through oncogenic factors such as β-Catenin and c-Myc to modulate the tumor immune milieu[2]. In addition, the stromal cells in tumors, i.e. cancer-associated fibroblasts (CAFs)[3,4], mediate an immunosuppressive activity by restricting the infiltration or activation of crucial immune cell lineages in the tumor microenvironment[5,6]. Growth factors, cytokines and chemokines, reactive oxygen species, and mechanical signaling have been shown to regulate the generation, maintenance, and activity of CAFs through various transcription

factors such as SMADs, STAT3, NF-κB, HSF1, RBP-J, YAP etc.[5]. Since targeting CAFs is considered a promising therapeutic approach, it is important to unveil other critical transcriptional regulatory mechanisms underlying CAF-mediated tumor progression.

The human prostate consists of 4 different anatomic zones, of which the peripheral zone (PZ) and transition zone (TZ) are the two major parts[7,8]. Interestingly, 70-80% of prostate cancers originate in the PZ, whereas only 20%–25% of prostate cancers begin in the TZ. In addition, the TZ tumors are often associated with favorable pathological features and better recurrence-free survival even though these tumors are often larger when diagnosed as compared to the PZ tumors because they are less readily detected by standard biopsy schemes due

[1]Department of Urology, University of Washington, Seattle, WA, USA. [2]Dan L. Duncan Comprehensive Cancer Center, Baylor College of Medicine, Houston, TX, USA. [3]Department of Biochemistry, University of Washington, Seattle, WA, USA. [4]Institute of Stem Cell and Regenerative Medicine, University of Washington, Seattle, WA, USA. [5]Department of Laboratory Medicine and Pathology, University of Washington, Seattle, WA, USA. [6]Division of Human Biology, Fred Hutchinson Cancer Research Center, Seattle, WA, USA. ✉e-mail: xin18@uw.edu

to their location[9,10]. These clinical observations raised an interesting yet unsubstantiated hypothesis that different tissue microenvironments in the two zones cause the distinct tumor frequency and malignant potential[8,11,12].

While investigating transcriptomic differences between human PZ and TZ stromal cells, we corroborated a previous report that *FOXF2*, a member of the forkhead box (FOX) transcription factor family, is expressed at a lower level in the PZ than in the TZ stroma[13]. Foxf2 is expressed in the mesenchyme of organs derived from primitive gut

and plays a crucial role in the development of these organs[14]. Stromal Foxf2 can induce *Sfrp1*[15] and *Bmp4*[16] to antagonize Wnt signaling in adjacent epithelia, repress *Fgf18* to downregulate Shh in adjacent epithelia[17], and promote[15,18] or antagonize[19] Tgfβ signaling. Importantly, decreased stromal Foxf2 has been shown to enhance epithelial proliferation during mouse gut development and promote adenoma formation by increasing the epithelial β-Catenin activity through stromal-epithelial interaction[15,20]. Therefore, we sought to investigate whether and how stromal Foxf2 affects prostate cancer progression.

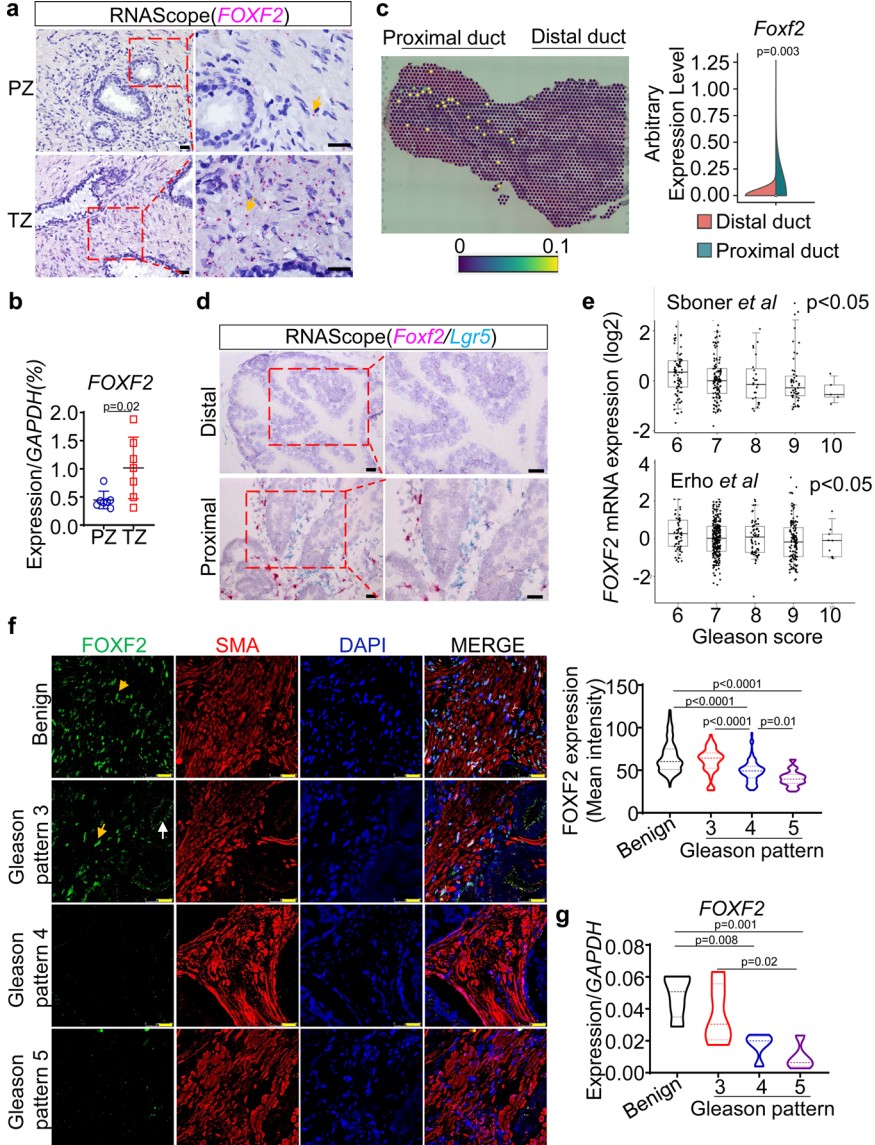

**Fig. 1 | Stromal FOXF2 expression inversely correlates with Gleason grade.**
**a** Representative image of 3 independent RNA-in-situ analysis of *FOXF2* in human transition (TZ) and peripheral zone (PZ) prostates. Arrows point to pink staining of *FOXF2*. Bars = 20 μm. **b** qRT-PCR of *FOXF2* in FACS-isolated Lin(CD45CD31CD235a)⁻Trop2⁻ human PZ and TZ prostate stromal cells. Data represent means ± s.d. from 7 specimens by two-sided unpaired t-test. **c** Image of Visium spatial gene expression analysis for *Foxf2* in prostate and urethra of 10-wk-old C57BL/6 mouse. Violin graph shows average arbitrary expression level of *Foxf2* in distal and proximal prostatic ducts. *N* = 4 mice. Two-sided unpaired t-test. **d** Representative image of 3 independent RNA-in-situ analysis of *Foxf2* and *Lgr5* in proximal and distal anterior prostate ducts of 10-wk-old C57BL/6 mouse. Bars = 25 μm. Three experiments were repeated independently with similar results. **e** *FOXF2* expression inversely correlates with Gleason scores in two human prostate cancer datasets: Erho et al. (*N* = 545) and Sboner et al. (*N* = 281). Box plots represent 5% (lower whisker), 25%

(lower box), 50% (median), 75% (upper box), and 95% (upper whisker). **f** Co-immunostaining of FOXF2 (AF6988, R&D) and α-smooth muscle actin (SMA) in primary human prostate cancer specimens. Yellow and white arrows point to nuclear staining of stromal FOXf2 and nonspecific apical staining in tumor cells, respectively. Dot plot shows relative fluorescence intensity of FOXF2 in benign prostate tissues and prostate cancers of different Gleason patterns, analyzed by one-way ANOVA with Turkey's multiple comparison test. Each dot represents value calculated from one field. Data show values collected from 343 fields from 20 prostate cancer specimens. Bars = 25 μm. **g** Expression of *FOXF2* in laser-captured stromal cells from prostate cancer of different Gleason patterns and adjacent benign tissues by qRT-PCR. Each dot in plot represents value calculated from one laser captured specimen. Data show values collected from 20 laser-captured samples from 13 prostate cancer specimens, analyzed by one-way ANOVA with Turkey's multiple comparison test. Source data are provided as a Source Data file.

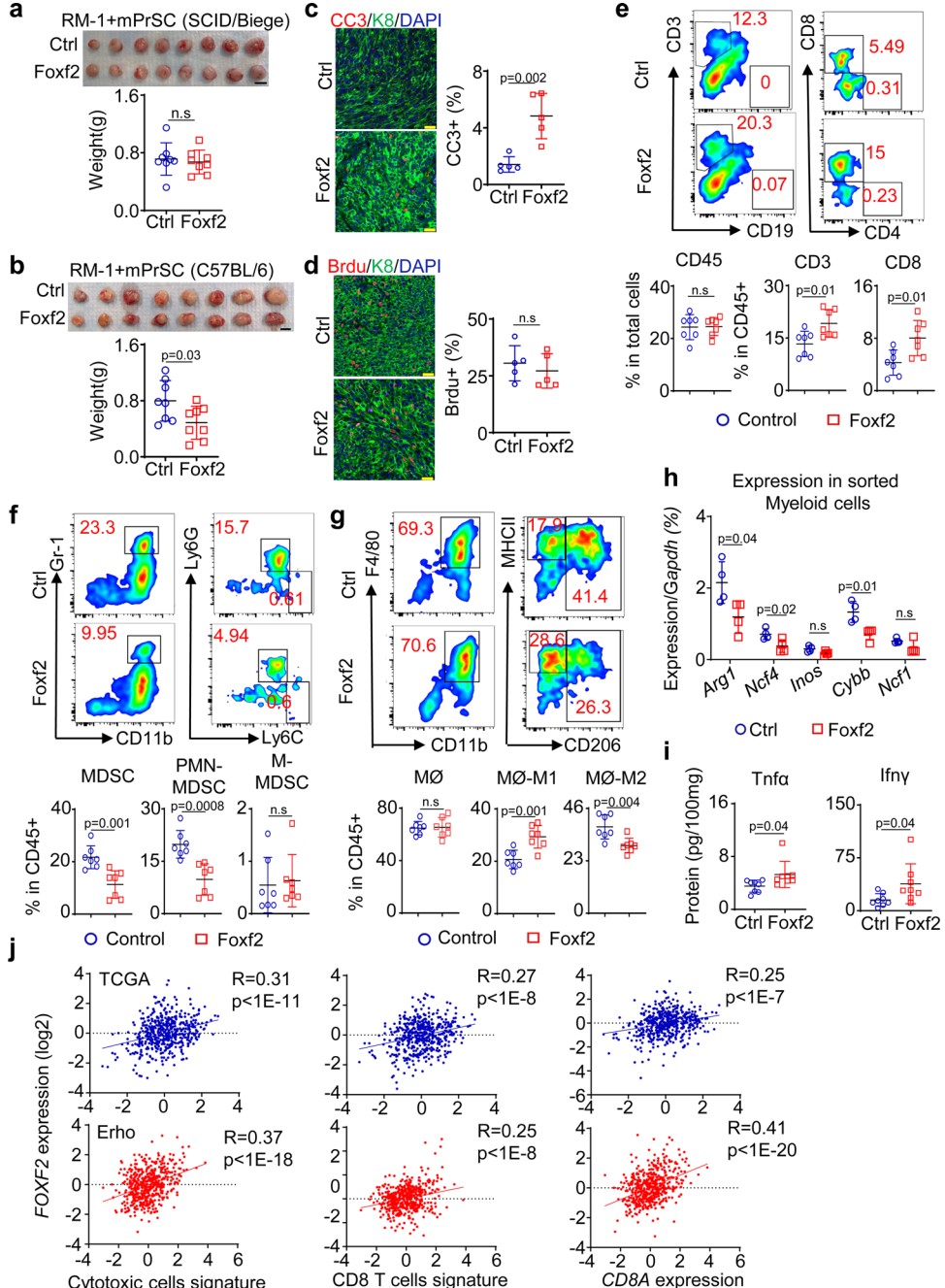

**Fig. 2 | Increasing stromal Foxf2 suppresses RM-1 growth in immunocompetent hosts. a** Representative image of RM-1 tumors grown subcutaneously with control (Ctrl) and Foxf2-expressing mouse prostate stromal cells in male SCID/Biege hosts. Dot plot shows means ± s.d. of tumor weight. Bar = 1 cm. *N* = 8 tumors per group. Two-sided unpaired t-test. **b** Representative image of RM-1 tumors grown subcutaneously with control and Foxf2-expressing mouse prostate stromal cells in male C57BL/6 hosts. Dot plot shows means ± s.d. of tumor weight. Bar = 1 cm. *N* = 8 tumors per group. Two-sided unpaired t-test. Co-immunostaining of Keratin 8 (K8) and cleaved caspase 3 (CC3) (**c**) and K8 and BrdU (**d**). Dot plots show means ± s.d. of CC3+ (**c**) and BrdU+ (**d**) cells from 5 tumors per group. Each dot represents average value from 20 fields per tumor. Bars = 25 μm. (e-g) FACS plots of T cells (**e**), myeloid derived suppressor cells (MDSC) (**f**), and macrophages (MØ) (**g**) in RM-1 tumors grown with control and Foxf2-expressing mouse prostate stromal cells. Dot plots show means ± s.d. of percentages of CD45+, CD3+ and CD8+ cells (**e**), monocytic (M-) and polymorphonuclear (PMN-) MDSC cells (**f**) and MHCII+CD206- M1 and CD206+ M2 cells (**g**). *N* = 7 tumors per group. two-sided unpaired t-test. **h** qRT-PCR of gene expression in CD45+CD11b+Gr-1+ myeloid cells FACS-sorted from RM-1 tumors grown with control and Foxf2-expressing stromal cells. Data represent means ± s.d. from 4 tumors per group. Two-sided unpaired t-test. **i** ELISA of Tnfα and Infγ in lysates of RM-1 tumors grown with control and Foxf2-expressing stromal cells. Data represent means ± s.d. from 8 tumors per group. Two-sided unpaired t-test. **j** Scatter plots show correlation between expression level of *FOXF2* and that of *CD8A* (left panels), CD8 T-cell signature (middle panels), and cytotoxic T-cell signature (right panels) in prostate cancer datasets by TCGA (upper panels, *N* = 498) and Erho et al. (lower panels, *N* = 546) by Pearson correlation analysis. Source data are provided as a Source Data file.

In this study, we take a combination of bioinformatic, molecular, cellular, and genetic approaches and reveal a mechanism in which stromal Foxf2 suppresses prostate cancer progression and metastasis by enhancing antitumor immunity.

## Results

### Stromal FOXF2 level inversely correlates with prostate cancer grade

Previous studies reported that *FOXF2* is highly expressed in the TZ prostate[21,22]. Our RNA-in-Situ analysis showed that *FOXF2* was expressed in the human prostate stromal but not epithelial cells, and that the expression level was higher in the TZ stroma (Fig. 1a). To further confirm the higher expression of *FOXF2* in the TZ stroma, we FACS-isolated the Lin(CD45CD31CD235a)⁻Trop2⁻ TZ and PZ stromal cells from 7 pathologist-verified fresh benign prostate tissues from prostate cancer patients who had undergone radical prostatectomy and examined the expression of *FOXF2* by qRT-PCR. Figure 1b shows that the expression level of *FOXF2* in TZ stroma was approximately 3-fold of that in the PZ stroma. Previously, we showed that the periurethral human TZ tissues share molecular features with the proximal mouse prostate ducts adjacent to the urethra[23,24]. A Visium spatial gene expression array revealed that *Foxf2* is also highly expressed in the mouse proximal prostate (Fig. 1c). We previously reported that the stromal cells in the mouse proximal prostate highly express *Axin2* and *Lgr5*[23,25]. An RNA-In-Situ duplex analysis of *Lgr5* and *Foxf2* further confirmed that *Foxf2* is specifically expressed by the stromal cells and the expression level is higher in the mouse proximal prostatic ducts than in distal ducts (Fig. 1d). This indicates that regulation of the expression pattern of *Foxf2* is conserved between human and mouse and further supports that the human and mouse prostates share anatomy-associated molecular features.

Bioinformatic analysis shows that the expression level of *FOXF2* inversely correlated with the Gleason scores of the primary prostate cancer specimens (Fig. 1e) in two independent human prostate cancer datasets[26,27]. Since the bioinformatic analysis cannot distinguish a lower expression level of *FOXF2* from a lower stromal composition in the tumors, we performed additional immunostaining. We evaluated several commercial antibodies against FOXF2. Two antibodies work well for Western blot (Supplementary Fig. 1a), whereas the specificity of only a sheep antibody can be confirmed in immunostaining using *Foxf2* knockout mouse embryos (Supplementary Fig. 1b). Unfortunately, this antibody stained apical membranes of cancer cells in human prostate cancer specimens (white arrows, Fig. 1f and Supplementary Fig. 1c). We reasoned that this staining was nonspecific because RNA-In-Situ analysis showed no sign of *FOXF2* in corresponding areas in sequential slides (Supplementary Fig. 1c). In contrast, the stromal cells exhibited expected nuclear staining with this antibody (yellow arrow, Fig. 1f). The intensity of the stromal nuclear staining correlated with the expression level of *FOXF2* determined by RNA-In-Situ in the stromal cells (Supplementary Fig. 1d), supporting that nuclear staining in the stroma was specific for FOXF2. Analysis of immunostaining intensity of stromal FOXF2 confirms the bioinformatic analysis and shows that the stromal FOXF2 expression level inversely correlates with the Gleason patterns (violin plot, Fig. 1f). We also determined the expression of *FOXF2* in laser-captured stromal and tumor cells from human prostate cancer specimens of different Gleason patterns by qRT-PCR. Our result confirms that the stromal expression of *FOXF2* inversely correlates with the Gleason patterns (Fig. 1g). In contrast, *FOXF2* is barely expressed in the prostate tumor cells (Supplementary Fig. 1e), which is consistent with the RNAScope analysis shown in Supplementary Fig. 1c. Collectively, these observations imply that stromal FOXF2 plays a potential role in prostate cancer progression.

### Stromal Foxf2 enhances antitumor immunity to inhibit prostate cancer

We used a lentivirus to overexpress Foxf2 in primarily cultured mouse prostate stromal cells (mPrSC) ($\Delta$Ct$^{(Foxf2\text{-}Gapdh)}$ = 4.50) and an immortalized human prostate stromal cell line WPMY-1 ($\Delta$Ct$^{(Foxf2\text{-}GAPDH)}$ = 4.75) (Supplementary Fig. 2a). The overexpression level was approximately 4-5-fold of that in the human TZ stromal cells ($\Delta$Ct$^{(FOXF2\text{-}GAPDH)}$ = 6.74). Overexpression did not significantly affect the growth of the stromal cells (Supplementary Fig. 2b). To determine whether increasing stromal Foxf2 expression affects prostate cancer cell growth in vitro, we cocultured the control and Foxf2-expressing mPrSC and WPMY-1 cells with mouse and human prostate cancer cells, respectively. The prostate cancer cell lines that we tested included two C57BL/6 syngeneic mouse prostate cancer cells [RM-1[28] and a Pten/Kras line that we established from the mouse model with prostate specific *Pten* deletion and *Kras*$^{G12D}$ activation[29] (Supplementary Fig. 2c)] and three human prostate cancer cell lines (PC3, DU145, and LnCaP cells). In vitro coculturing of the RM-1 cells and Foxf2-overexpressing stromal cells shows that stromal Foxf2 did not affect the migration or invasion of RM-1 cells in a paracrine manner (Supplementary Fig. 2d).

The Foxf2-expressing WPMY-1 cells did not affect the subcutaneous growth of PC3 or LnCaP cells in vivo (Supplementary Fig. 2e, f) in male SCID/Beige host mice. Nor did the Foxf2-expressing mPrSC affect the subcutaneous growth of RM-1 cells in male SCID/Beige host mice (Fig. 2a). There was also no significant change in angiogenesis (Supplementary Fig. 2g), or the expression of genes associated with epithelial-mesenchymal transition (Supplementary Fig. 2h) and cancer stem cells (Supplementary Fig. 2i) in the RM-1 tumor xenografts. In contrast, Foxf2-expressing mPrSC suppressed the subcutaneous growth of RM-1 cells by 31% in the C57BL/6 male hosts (Fig. 2b). Immunostaining analysis showed a 3.4-fold increase in the apoptotic index in the Foxf2 overexpression group (Fig. 2c) but no significant difference in the proliferation index (Fig. 2d). Similarly, the Foxf2-expressing mPrSC cells only suppressed the subcutaneous growth of the Pten/Kras cells in the C57BL/6 (Supplementary Fig. 2j) but not in SCID/Beige hosts (Supplementary Fig. 2k). These results indicate that stromal Foxf2 imposes a tumor suppressive effect by modulating the immune system.

We examined the immune cell composition within the RM-1 xenografts by flow cytometry. Tumors of similar size were analyzed to exclude the impact of tumor size on immune cell composition. *Foxf2* overexpression did not alter the ratio of the CD45$^+$ leukocytes within the tumors but increased the CD3$^+$ T cells (mostly CD8$^+$ cells) by 1.5-fold (Fig. 2e). There was a 50% decrease in the CD11b$^+$Gr-1$^+$ myeloid derived suppressor cells (MDSC) (mainly Ly6G$^+$ PMN-MDSC) in the Foxf2-expressing stromal group (Fig. 2f). Foxf2 overexpression did not alter the ratio of the CD11b$^+$F4/80$^+$ macrophages but shifted the polarity of the macrophages from a relatively immunosuppressive M2 state (CD11b$^+$F4/80$^+$CD206$^+$) to a more inflammatory M1 state (CD11b$^+$F4/80$^+$CD206$^-$MHCII$^+$) (Fig. 2g). Stromal Foxf2 overexpression downregulated the immunosuppression-related genes in MDSCs including *Arg1*, *Ncf4*, and *Cybb* (Fig. 2h). MDSCs were also slightly decreased in the spleens of host mice in the Foxf2 group (Supplementary Fig. 3a) whereas no change was noted in the blood of host mice (Supplementary Fig. 3b). ELISA showed higher expressions of Tnfα and Ifnγ in the tumor lysates of the Foxf2-expressing group (Fig. 2i), indicating an enhanced T-cell activity. Similar changes in the immune cell lineages were also noted in the Pten/Kras syngeneic model (Supplementary Fig. 3c). Conversely, suppressing Foxf2 in mPrSC cells promoted the growth of cocultured RM-1 cells in vivo (Supplementary Fig. 3d) and induced an immunosuppressive tumor microenvironment (Supplementary Fig. 3e). In summary, these results support that stromal Foxf2 enhances antitumor immunity.

Although overexpressing Foxf2 in mPrSC did not suppress the growth of RM-1 cells in the SCID/Beige hosts (Fig. 2a), it also reduced

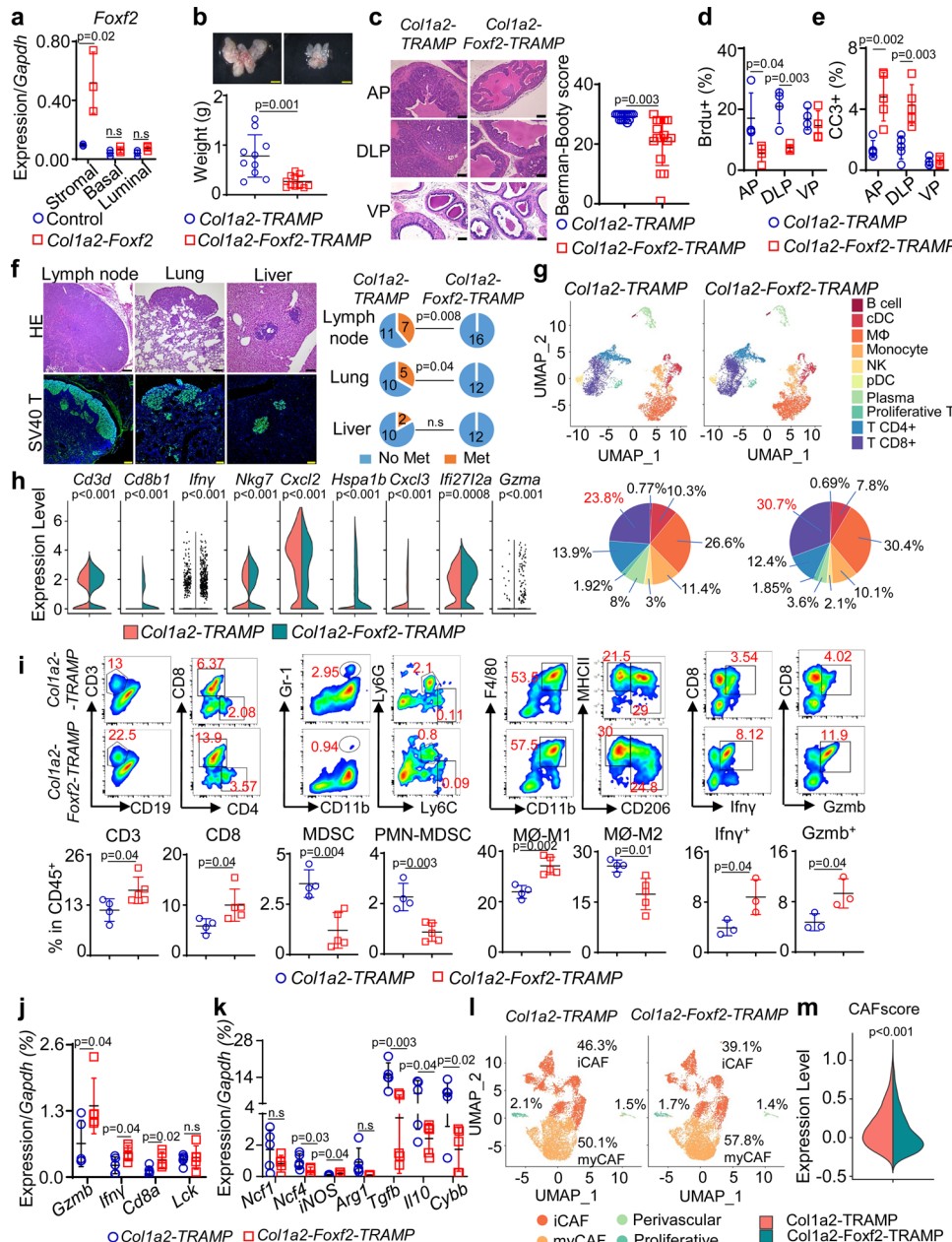

**Fig. 3 | Increasing stromal Foxf2 expression suppresses tumor progression and metastasis in TRAMP mice. a** qRT-PCR of *Foxf2* using cells from tamoxifen-treated 10-wk-old mice (*N* = 3). Dot plot shows means ± s.d. Two-sided unpaired t-test. **b** Images of prostates from 9-mth-old *Col1a2-TRAMP* (*N* = 10) and *Col1a2-Foxf2-TRAMP* (*N* = 11) mice. Dot plot shows means ± s.d. Two-sided unpaired t-test. Bars = 2 mm. **c** H&E staining of anterior (AP), dorsolateral (DLP), and ventral (VP) prostates of 9-mth-old mice. Bars = 75 μm. Dot plot shows Berman-Booty scores of histology. Dots represent scores calculated from 16 tissue slides per group from 7 *Col1a2-TRAMP* and 6 *Col1a2-Foxf2-TRAMP* mice. Unpaired t-test. Dot plots show means ± s.d. of BrdU⁺ (**d**) and CC3⁺ (cleaved caspase 3) (**e**) cells in AP, DLP, and VP of 9-mth-old mice. *N* = 4 for *Col1a2-TRAMP*; *N* = 5 for *Col1a2-Foxf2-TRAMP*. Dots represent mean values from 20 representative fields of individual mice. **f** H&E and immunostaining of SV40. Pie charts show percentage of mice with metastasis from a total of 12–18 mice. Bars = 100μm. **g** UMAP plot of 13,948 FACS-isolated CD45⁺ cells in prostates of three 9-mth-old mice identifies 10 subpopulations annotated manually based on expression of genes shown in Supplementary Fig. 7a. Pie charts show

percentages of individual subpopulations. **h** Violin plots show differences of gene expressions with a Wilcoxon Rank Sum test used within Seurat's "FindAllMarkers" function. **i** FACS plots of immune cell lineages in anterior prostate cancer tissues of 9-mth-old mice. Dot plots show means ± s.d. of percentages of individual cell lineages. For analyses of T, MDSC and Macrophages, *N* = 4 mice for *Col1a2-TRAMP*; *N* = 5 mice for *Col1a2-Foxf2-TRAMP*. For analyses of Infγ and Gzmb, *N* = 3 mice for both groups. Two-sided unpaired t-test. MDSC: myeloid derived suppressor cells. (**j, k**) qRT-PCR using FACS-isolated CD45⁺CD3⁺ T and CD45⁺CD11b⁺ myeloid cells from prostates of 9-mth-old mice. Dot plots show means ± s.d. of gene expression. *N* = 5 mice per group. Two-sided unpaired t-test. **l** UMAP plot of 18,000 FACS-isolated Lin⁻CD49f⁺CD24⁻ prostate stromal cells from 9-mth-old mice identifies 4 cancer-associated fibroblast (CAF) subpopulations annotated manually based on expression of genes presented in Supplementary Fig. 7e. **m** Violin plot shows mean CAF score calculated using 30 CAF-associated genes in Seurat's AddModuleScore function in 9-mth-old mice. Two-sided Wilcoxon Rank Sum test was used to identify differential gene expression. Source data are provided as a Source Data file.

the frequency of immunosuppressive myeloid cells (Supplementary Fig. 3f). This suggests that the changes in the myeloid lineage did not directly suppress the tumor growth. Nor did the Foxf2-expressing mPrSC affect the subcutaneous growth of RM-1 cells in male nude host mice that have functioning natural killer (NK) cells (Supplementary Fig. 3g). This excludes the contribution of NK cells to the antitumor effect of Foxf2. Collectively, these results indicate that a higher stromal Foxf2 expression suppresses tumor growth by enhancing the CD8⁺ T-cell infiltration and activity. To extend this finding in human prostate cancer, we took an unbiased bioinformatic approach to investigate the correlation between *FOXF2* expression and cytotoxic T cells. The expression level of *FOXF2* positively correlate with that of *CD8A* in both prostate cancer datasets by TCGA and Erho et al.[26] (Fig. 2j). In addition, a CD8 T-cell signature and a cytotoxic T-cell signature[30] also positively correlate with the expression of *FOXF2* in both datasets (Fig. 2j).

## Stromal Foxf2 affects TIME and CAF phenotype in TRAMP

We sought to corroborate these findings in a mouse model for autochthonous prostate cancer. Although Foxf2 is differentially expressed between the stromal cells at the mouse proximal and distal prostatic ducts, it is technically infeasible to evaluate the impact of this differential expression pattern on tumor initiation and progression. This is because the proximal ducts only constitute a very small region within the mouse prostate and the stromal cells are much less abundant in the mouse prostate than in the human prostate[31]. To overcome the limitation, we generated an *R26-LSL-Foxf2* mouse model in which *Foxf2* was knocked into the *ROSA26* locus downstream of a floxed transcriptional stop signal (LSL) (Supplementary Fig. 4a) and bred this line with the *Col1a2-CreER^{T2}* mouse line[32] to achieve temporal *Foxf2* upregulation in the prostate stromal cells. Treating the *Col1a2-CreER^{T2};R26-LSL-Foxf2^{Tg/Tg}* bigenic mice (hereafter referred to as *Col1a2-Foxf2*) with tamoxifen successfully and specifically upregulated *Foxf2* in the prostate stromal cells to approximately 5.5-fold of the endogenous level (Fig. 3a). Supplementary Fig. 4b further confirms that Foxf2 was upregulated at the protein level in the FACS-isolated prostate stromal cells.

We employed the Transgenic Adenocarcinoma of the Mouse Prostate (TRAMP) mouse model for prostate cancer to determine whether stromal Foxf2 affected tumor progression. This model expresses the SV40 T antigen driven by a prostate specific promoter[33]. On the C57BL/6 background, it develops prostatic intraepithelial neoplasia (PIN) at 3 months and prostate cancer with focal neuroendocrine differentiation (particularly in ventral prostate) at 6 months. We generated a cohort of *Col1a2-CreER^{T2};R26-LSL-Foxf^{Tg/Tg};TRAMP^{Wt/Mut}* (*Col1a2-Foxf2-TRAMP*) and *Col1a2-CreER^{T2};TRAMP^{Wt/Mut}* (*Col1a2-TRAMP*) mice. Eight-week-old *Col1a2-Foxf2-TRAMP* mice were treated with tamoxifen to activate stromal Foxf2 expression. Age-matched *Col1a2-TRAMP* littermates were treated the same way and served as the controls. Figure 3b shows that the size and weight of the prostates of *Col1a2-Foxf2-TRAMP* mice were significantly reduced compared to those of the control mice at 9 months. The reduction was mainly in the anterior (AP) and dorsolateral (DLP) lobes but not the ventral (VP) lobes (Supplementary Fig. 4c). H&E staining shows that the control (*Col1a2-TRAMP*) mouse prostates uniformly and diffusely displayed invasive adenocarcinomas in the dorsal and anterior lobes, whereas *Col1a2-Foxf2-TRAMP* mouse prostates showed a spectrum of less advanced lesions in the same lobes (Fig. 3c). We evaluated and scored the histology of the tumors based on the published Bernam-Booty scoring system[34]. Median adjusted scores of the control and *Col1a2-Foxf2-TRAMP* mouse prostate cancers were 30 and 22, respectively and were statistically different (Fig. 3c), supporting a delayed disease progression in the *Col1a2-Foxf2-TRAMP* group. Immunostaining shows that the expression patterns of SV40 T and prostate lineage markers (the androgen receptor, Krt5, and Krt8) were the same between the groups (Supplementary Fig. 4d). Consistent with the delayed tumor progression, the proliferating index of tumor cells was

reduced as determined by immunostaining of BrdU in AP and DLP (Fig. 3d). Immunostaining of the cleaved caspase 3 also indicates an increased apoptosis in AP and DLP (Fig. 3e). More importantly, distal metastases at the lumbar lymph nodes and lung, but not liver, were significantly reduced in the *Col1a2-Foxf2-TRAMP* mice as compared to the control mice (Fig. 3f). These results further corroborate that increased stromal Foxf2 suppresses prostate cancer progression in an immunocompetent background.

In our C57BL/6 colonies, approximately 46% of TRAMP mice developed neuroendocrine prostate cancer (NEPC) at the 9 months of age. NEPCs in ventral prostate are likely developed de novo and tends to form focal and huge tumors (Supplementary Fig. 5a), whereas NEPCs in anterior and dorsolateral lobes are relatively rare and sporadic. We evaluated whether stromal Foxf2 expression impacted the development of NEPC. Immunostaining of the NEPC marker synaptophysin indicates that there was no difference in the frequency of NEPC in the ventral prostate of experimental mice in the control and Foxf2-expressing groups (Supplementary Fig. 5a). Immunostaining of synaptophysin and qRT-PCR analysis of the NEPC-related genes (*Syp, Chga, and Eno2*) in the adenocarcinoma of different lobes also showed no significant changes in NEPC cell frequency (Supplementary Fig. 5b) or NEPC gene expression (Supplementary Fig. 5c), respectively. This data indicate that stromal Foxf2 expression does not affect de novo development of NEPC or neuroendocrine trans-differentiation in this model.

Another potential mechanism for the delayed tumor progression in the TRAMP model is through delayed peripheral tolerance towards the SV40 Tag-IV antigen. In our experiments, when stromal Foxf2 expression was activated, the mice were already around 9-10 weeks of age. At this age, the TRAMP model has been reported to develop complete tolerance[35]. Nevertheless, to directly exclude this mechanism, we treated *TRAMP* and *TRAMP;Col1a2-CreER^{T2};R26-LSL-Foxf2* mice with tamoxifen at 8 weeks of age to activate stromal Foxf2 expression and immunized them at 11 weeks of age (young) and 6 months of age (old) with SV40 Tag-IV-pulsed dendritic cells. Splenocytes from these mice were stimulated with SV40 Tag-IV in vitro. Supplementary Fig. 6 shows that the CD8⁺ T cells from the two groups responded similarly as determined by the production of INFγ. This result demonstrates that stromal Foxf2 expression does not impact the tolerance of TRAMP model toward Tag-IV.

We performed single-cell RNA-Seq of FACS-isolated CD45⁺ leukocytes from both groups. Unsupervised clustering analysis on integrated single-cell datasets revealed an increased CD8⁺ T-cell frequency (Fig. 3g and Supplementary Fig. 7a, 7b) and activity and a decreased MDSC activity (Fig. 3h) in the *Col1a2-Foxf2-TRAMP* mice. Flow cytometric analysis of anterior prostate cells confirmed the increased CD8⁺ T cells, decreased MDSCs (mainly PMN-MDSC), and a shift in macrophage polarity toward the proinflammatory (M1) state (Fig. 3i). The T-cell activity was enhanced as demonstrated by the increased numbers of Ifnγ- and Gzmb-expressing cells (Fig. 3i) and upregulation of the genes related to T-cell infiltration and antitumor immunity (Fig. 3j). In addition, the expression of 7 genes related to the immunosuppressing activity of the myeloid cells were decreased (Fig. 3k). These changes were also noted in DLP (Supplementary Fig. 7c) but not in VP (Supplementary Fig. 7d). It remains unclear why tumor growth and immune microenvironment in VP was not significantly affected.

To determine the impact of Foxf2 on prostate stromal cell biology, we also performed an scRNA-Seq analysis of the Lin⁻CD49f⁻CD24⁻ prostate stromal cells from both groups. The analysis in TRAMP mice revealed two major subpopulations that represented the myofibroblastic CAF (myCAF) and inflammatory CAF (iCAF) and two small subpopulations that corresponded to the perivascular and proliferative stromal cells (Fig. 3l and Supplementary Fig. 7e, 7f). The percentage of myCAF (57.8%) in the *Col1a2-Foxf2-TRAMP* mice is higher than that of the control mice (50.1%) (Fig. 3l), suggesting that Foxf2 induced a shift toward the myCAF phenotype. On the other hand, the

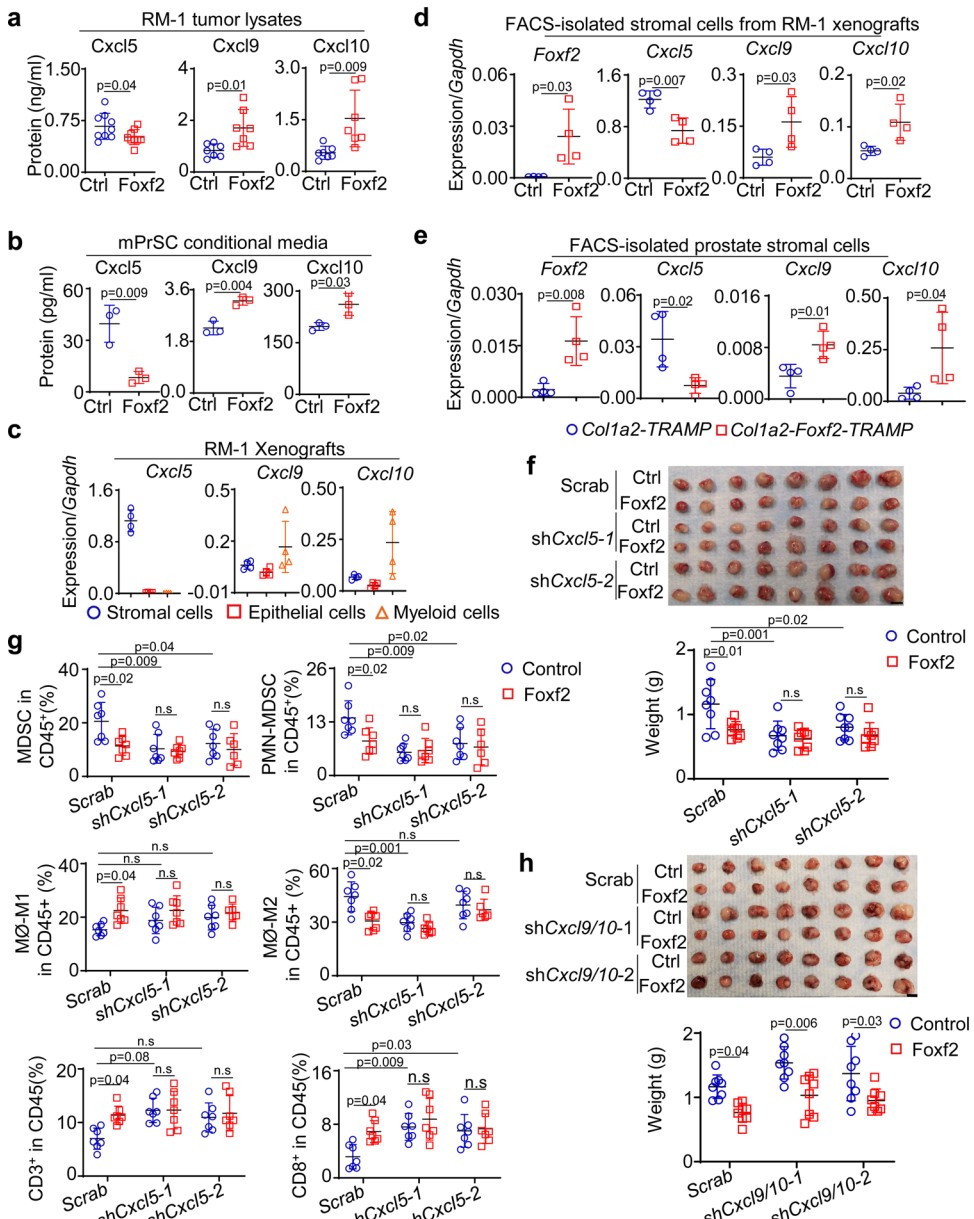

**Fig. 4 | Cxcl5 plays a crucial role in stromal Foxf2-mediated tumor suppression.** ELISA of Cxcl5, Cxcl9 and Cxcl10 in lysates of RM-1 tumors grown with control (Ctrl) and Foxf2-expressing mouse stromal cells (**a**) and supernatant of in vitro cultured control and Foxf2-expressing mouse prostate stromal cells (mPrSC) (**b**). Data represent means ± s.d. of protein expression. In Fig. 4, *N* = 9 per group for Cxcl5, *N* = 7 per group for Cxcl9 and Cxcl10. In Fig. 4b, *N* = 3 per group. Two-sided unpaired t-test. **c** qRT-PCR of *Cxcl5/9/10* in FACS-isolated stromal, RM-1, and myeloid cells from RM-1 tumors grown with mouse prostate stromal cells. Data represent means ± s.d. of gene expression from 4 tumors per group. **d**, **e** qRT-PCR of *Foxf2* and *Cxcl5/9/10* in FACS-isolated stromal cells from RM-1 tumors grown with control and Foxf2-expressing stromal cells and from prostates of 9-mth-old control (*Col1a2-TRAMP*) and *Col1a2-Foxf2-TRAMP* mice. Dot plots show means ± s.d. of gene

expression from 4 mice per group. **f** Image of RM-1 tumors grown with control and Foxf2-expressing stromal cells with scrambled shRNA or shRNAs against *Cxcl5*. Dot plot shows means ± s.d. of tumor weight. *N* = 8 per group. Two-way ANOVA with Tukey's multiple comparison test. Bar = 1 cm. **g** Dot plots show means ± s.d. of percentages of immune cell lineages determined by flow cytometry. *N* = 7 per group except that *N* = 6 for analysis of macrophages in *shCxcl5-2* group. Two-way ANOVA with Tukey's multiple comparison test. **h** Image of RM-1 tumors grown with control and Foxf2-expressing stromal cells with scrambled shRNA or shRNAs against *Cxcl9* and *Cxcl10*. Dot plot shows means ± s.d. of tumor weight. *N* = 8 per group. Two-way ANOVA with Tukey's multiple comparison test. Bar = 1 cm. Source data are provided as a Source Data file.

overall CAF gene signature score defined by the average expression of 30 CAF-associated genes (*S100a4, Acta2, Zeb1, Slc16a4, Pdpn, Foxf1, Fap, Vim, Pdgfrb, Sparc, Pdgfra, Mmp2, Mmp11, Aspn, Fn1, Mfap5, Ogn, Tnc, Col3a1, Col11a1, Col1a1, Col1a2, Emilin1, Col5a1, Col16a1, Loxl1, Thy1, Ly6a, Il-6, Has1*) on a single-cell level was slightly but significantly reduced in the *Col1a2-Foxf2-TRAMP* mice (Fig. 3m). The suppression of the CAF phenotype by Foxf2 was underestimated because many Foxf2-expressing stromal cells were probably assigned to the none-

expressing group due to the limitation of the genes detected per cell. Supplementary Fig. 7g shows downregulation of representative CAF-related genes in the prostate stromal cells of *Col1a2-Foxf2-TRAMP* mice.

**Cxcl5 is essential for stromal Foxf2-mediated tumor suppression**
Cytokines play crucial roles in tumor immunity. Stromal cells are the major source of many cytokines and chemokines in the prostate[36,37].

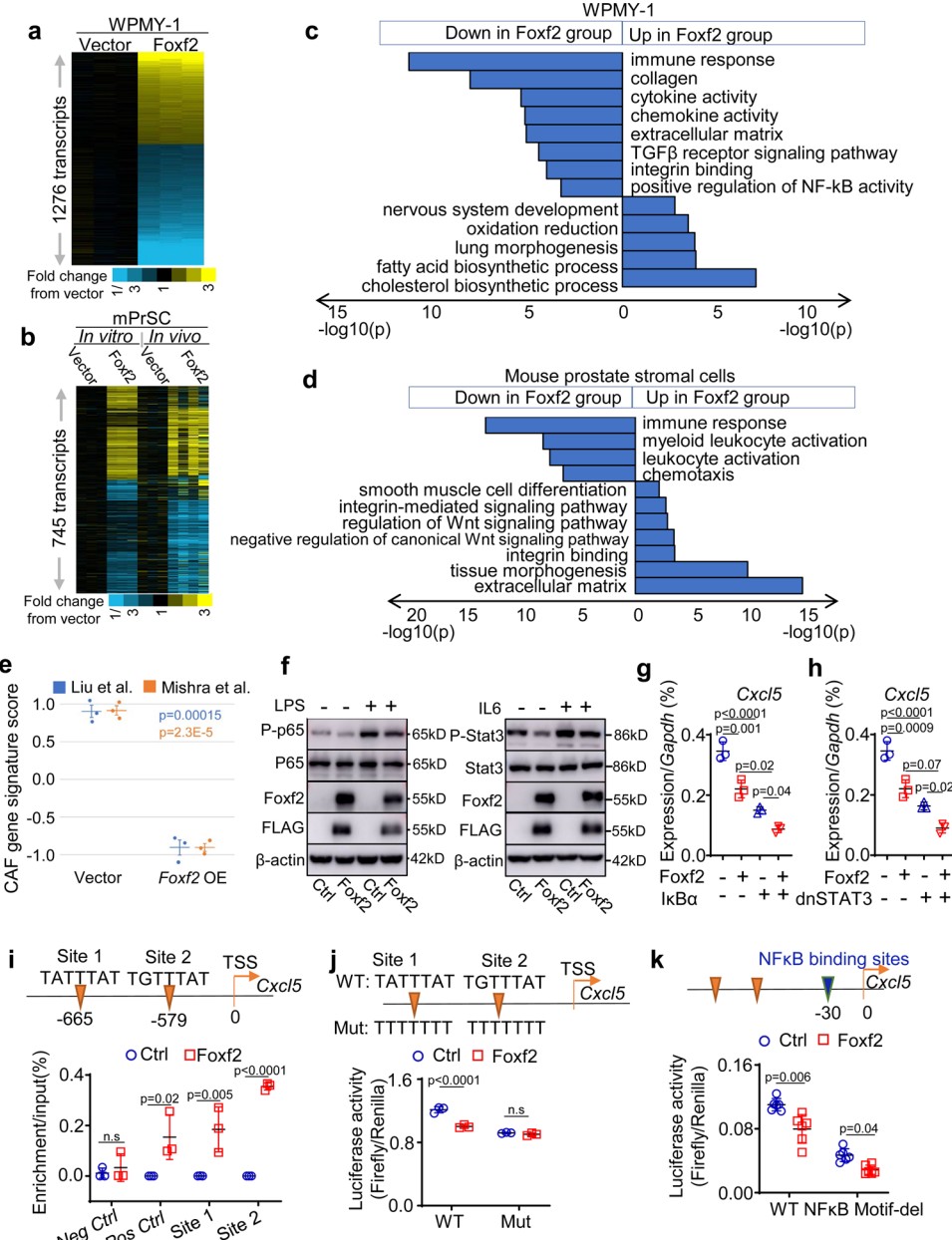

**Fig. 5 | Foxf2 suppresses Cxcl5 directly and indirectly.** Heatmap of RNA-Seq analysis of control and Foxf2-expressing WPMY-1 (*N* = 3 each) (**a**), in vitro cultured control and Foxf2-expressing mouse prostate stromal cells (*N* = 3 each) (**b**), and FACS-isolated control (*N* = 3) and Foxf2-expressing (*N* = 4) mouse prostate stromal cells from in vivo RM-1 tumors (**b**). Heatmaps depict fold changes in experiment versus control. Each gene is centered on average of control. **c**, **d** Gene ontology analyses of RNA-Seq analyses. **e** CAF gene signature scoring comparing Foxf2 overexpression (OE) with control WPMY-1 cells. Bars represent standard error. P-values by two-sided t-test. **f** Western blot analysis using control and Foxf2-expressing mouse prostate stromal cells. LPS: lipopolysaccharide. Experiments were repeated for three times with similar observation. qRT-PCR of *Cxcl5* in control and Foxf2-expressing mouse prostate

stromal cells transduced with and without super IκBα (**g**) or dominant negative STAT3 (dnSTAT3) (**h**). Dot plots show means ± s.d. of *Cxcl5* expression from 3 independent experiments. Two-way ANOVA with Tukey's multiple comparison test (**i**) ChIP analysis of Foxf2 binding at promoter of *Cxcl5* in mouse prostate stromal cells. TSS: transcription start site. Loci at Ch.15 and *Pdgfrα* serve as negative and positive binding controls, respectively. Dot plot shows means ± s.d. of relative enrichment from 3 independent experiments. Two-sided unpaired t-test. Luciferase reporter assays determine activity of *Cxcl5* reporter with and without mutations in putative Foxf2 (**j**) or NF-κB (**k**) binding sites in control and Foxf2-expressing mouse prostate stromal cells. Data represent means ± s.d. from 3 (**j**)−6(**k**) experiments. Two-sided unpaired t-test. Source data are provided as a Source Data file.

We reasoned that Foxf2 regulates stromal production of cytokines and chemokines, thereby altering the tumor immune microenvironment. We audited the expression of 31 common cytokines and chemokines in the RM-1 tumor lysates of the control and Foxf2-expressing stromal groups shown in Fig. 2b using a small-scale ELISA based cytokine assay. Supplementary Table 1 shows the 23 cytokines that were significantly altered. To identify the cytokines and chemokines that were directly regulated by Foxf2 in the stromal cells, we performed the same assay

using the in vitro cultured control and Foxf2-expressing mPrSC cells (Supplementary Table 1). Five cytokines and chemokines were differentially expressed at the same trend in the two analyses: Cxcl5 and Il-6 were downregulated while Cxcl9, Cxcl10, and Tnfα were upregulated in the *Foxf2*-expressing groups.

We focused on Cxcl5 and Cxcl9/10 because they have been shown to regulate recruitment and activity of the myeloid and T cells, respectively[38,39]. Figure 4a, b corroborates that Cxcl5 was decreased

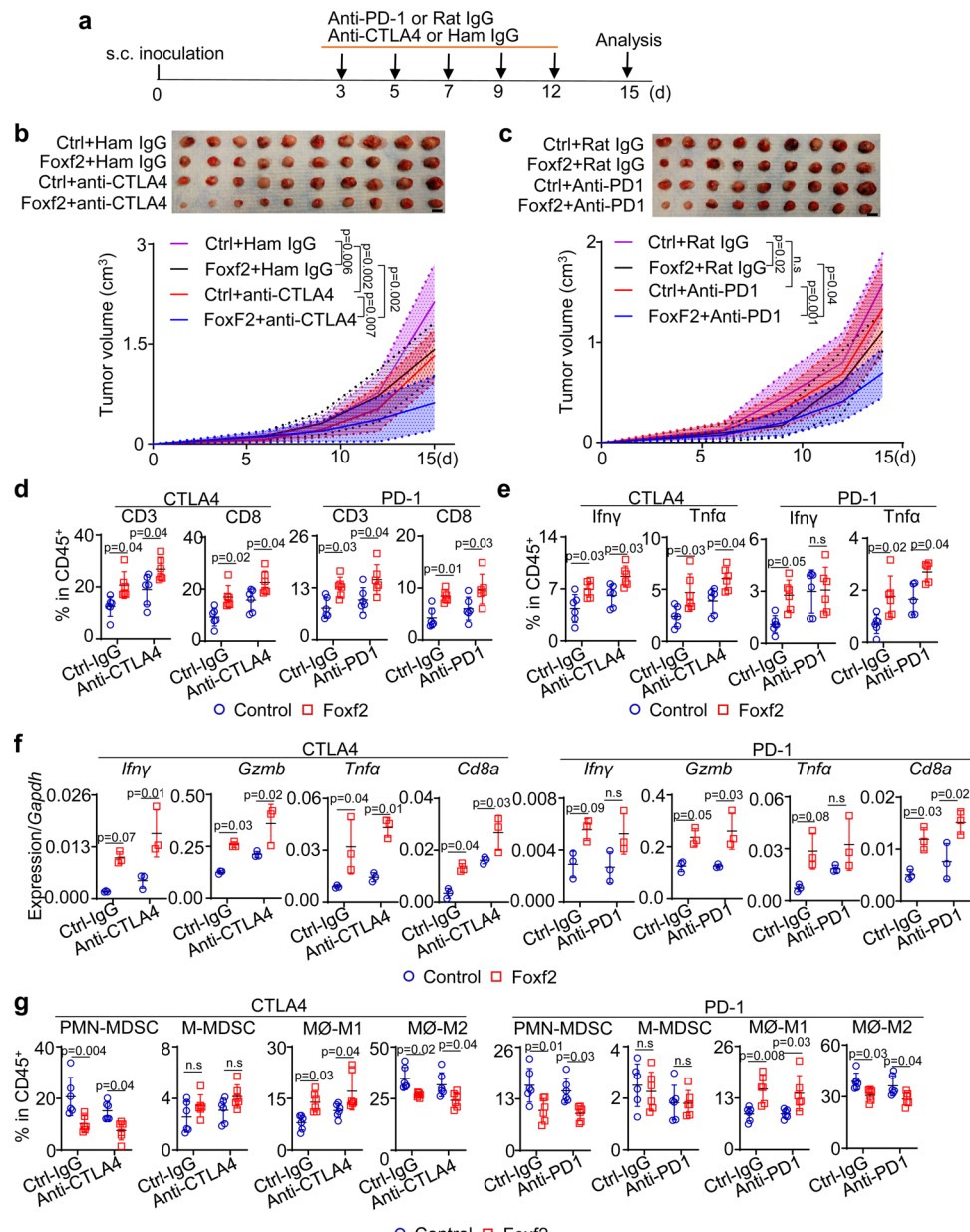

**Fig. 6 | Increasing Foxf2 sensitizes RM-1 to immune checkpoint inhibitors.**
**a** Schematic illustration of experimental design. Images of RM-1 tumors grown subcutaneously with control and Foxf2-expressing mouse prostate stromal cells with and without treatment of anti-CTLA4 (**b**) ($N = 10$) or anti-PD1 (**c**) ($N = 9$), Bar = 1 cm. Line charts show means ± s.d. of tumor volumes. Two-way ANOVA with Tukey's multiple comparison test. FACS analysis of T cells (**d**) and T cells expressing Ifnγ or Gzmb (**e**) in RM-1 tumors grown with control and Foxf2-expressing stromal cells with and without treatment by anti-CTLA4 or anti-PD1. Data represent means ± s.d. from 6 independent tumors, except that $N = 5$ for analysis of Tnfα in anti-PD1

group in Fig. 6e. Two-way ANOVA with Tukey's multiple comparison test. **f** qRT-PCR of gene expression in FACS-isolated CD3[+] T cells from RM-1 tumors in **b** and **c**. Data represent means ± s.d. from 3 specimens. Two-way ANOVA with Tukey's multiple comparison test. **g** FACS analysis of monocytic (M-) and polymorphonuclear (PMN-) myeloid derived suppressor cells (MDSC) and MHCII[+]CD206[-] M1 and CD206[+] M2 macrophage in RM-1 tumors grown with control and Foxf2-expressing stromal cells with and without treatment by anti-CTLA4 or anti-PD1. Data represent means ± s.d. from 6 independent tumors. Two-way ANOVA with Tukey's multiple comparison test. Source data are provided as a Source Data file.

while Cxcl9 and Cxcl10 were increased in the RM-1 tumor lysates of the Foxf2-expressing group and in the conditional media of Foxf2-expressing mPrSC cells. To further verify the cellular source of these cytokines in the RM-1 xenografts in vivo, we FACS separated prostate tumor epithelial cells (Lin[-]CD49f[+]), stromal cells (Lin[-]CD49f[-]) and myeloid cells (CD45[+]CD11b[+]) and examined the expression of the three cytokines by qRT-PCR. Figure 4c shows that the stromal cells were the major source of *Cxcl5* whereas *Cxcl9* and *Cxcl10* were expressed the highest by the myeloid cells. We further confirmed that *Cxcl5* was downregulated, and *Cxcl9/10* were upregulated in the FACS-isolated stromal cells from both RM-1 xenografts (Fig. 4d) and prostate cancer

tissues of *Col1a2-Foxf2-TRAMP* mice (Fig. 4e) as compared to their respective controls. Conversely, suppressing Foxf2 in mouse prostate stromal cells upregulated Cxcl5 (Supplementary Fig. 8a)

The myeloid cells in the RM-1 tumor xenografts expressed *Cxcr2*, the Cxcl5 receptor, at the highest level (Supplementary Fig. 8b). To determine whether Cxcl5 is a major player downstream of Foxf2, we investigated whether suppressing Cxcl5 can attenuate or ablate stromal Foxf2-mediated tumor suppressive effect. We cloned two *Cxcl5* shRNAs into the Foxf2-expressing lentiviral vector separately so that the lentiviruses expressed *Foxf2* at a physiologically relevant level ($\Delta Ct^{(Foxf2-Gapdh)} = 2.5$) and simultaneously suppressed *Cxcl5* in mPrSC

cells (Supplementary Fig. 8c). Briefly, mPrSC cells were infected by the lentivirus expressing the control scramble shRNA only, *Foxf2* and the scrambled shRNA, the *Cxcl5* shRNA1/2 only, and *Foxf2* and the *Cxcl5* shRNA1/2, respectively. RM-1 cells were mixed with each group of the engineered stromal cells separately and incubated subcutaneously in C57BL/6 male hosts.

Stromal Foxf2 expression suppressed the growth of RM-1 tumor in C57BL/6 hosts as expected (Fig. 4f). The *Cxcl5* shRNA-expressing mPrSC cells suppressed RM-1 growth than the scramble shRNA-expressing cells, which is expected because knocking down *Cxcl5* impairs MDSC recruitment and attenuates tumor growth. The knockdown of *Cxcl5* was confirmed by ELISA (Supplementary Fig. 4d) and the reduction of the immunosuppressive myeloid cells were confirmed by FACS (Fig. 4g). Importantly, Foxf2-expressing mPrSC cells did not significantly affect RM-1 growth when *Cxcl5* was knocked down. Nor was the immune cell composition significantly different among these groups (Fig. 4g). *Cxcl5* knockdown was capable of increasing T cells to a comparable level induced by increasing stromal *Foxf2*. These results demonstrate that Cxcl5 is a critical player in the stromal Foxf2-mediated tumor suppression.

Cxcl9/10 are important regulators of T-cell recruitment and activation[40]. They are both upregulated in the Foxf2-expressing mPrSC (Fig. 4a, b, and Supplementary Table 1). We investigated whether upregulation of stromal Cxcl9/10 also played a necessary role in the Foxf2-mediated tumor suppression. We generated lentiviral vectors that enable simultaneous suppression of *Cxcl9/10* (Supplementary Fig. 8e). Knocking down stromal *Cxcl9/10* did not ablate the Foxf2-mediated growth suppression of RM-1 tumors in C57BL/6 hosts (Fig. 4h) or ablate the changes in the T and myeloid lineages (Supplementary Fig. 8f, g). This result is not surprising because the myeloid cells but not the stromal cells are the major source of Cxcl9/10 in the RM-1 syngeneic model (Fig. 4c). Collectively, we show that Cxcl5, but not Cxcl9/10 from the stromal cells, is the major downstream effector of stromal Foxf2 in suppressing prostate tumor growth.

## Foxf2 transcriptionally represses *Cxcl5* directly and indirectly

To dissect the molecular mechanisms underlying the stromal Foxf2-mediated tumor suppression, we compared gene expression profiles of the control and Foxf2-expressing WPMY-1 and mPrSC cells by RNA-Seq. To take into consideration of gene regulation by tumor-stromal cell interaction, we also compared gene expression profiles of the control and Foxf2-expressing mPrSC cells FACS-isolated from the RM-1 xenografts shown in Fig. 2b. 1276 and 745 genes were differentially expressed by at least 1.4-fold between the control and Foxf2-overexpression group in WPMY-1 (Fig. 5a) and mPrSC cells (Fig. 5b), respectively. Gene Ontology analysis revealed that the genes associated with smooth muscle cell differentiation, tissue morphogenesis, TGFβR signaling pathway, and Wnt signaling pathway were altered, consistent with previous reports regarding regulation of these signaling pathways by Foxf2. Interestingly, genes associated with immune response, chemokine and cytokine activity, leukocyte activation were significantly downregulated in the Foxf2-expressing groups in both analyses (Figs. 5c, d). We scored CAF gene signatures based on two previous publications[41,42] in the Foxf2-expressing and control groups. For both signatures, the Foxf2-expressing WPMY-1 cells scored significantly lower than the control cells (Fig. 5e). The CAF gene signature score was not statistically significantly decreased in the Foxf2-expressing mouse prostate stromal cells, but some major CAF-associated genes were significantly downregulated including *Thy1*, *Ly6a*, *Pdgfrα*, and *Pdpn* etc. (Supplementary Fig. 9). These results further support the observation from the scRNA-seq analysis that Foxf2 moderately suppresses the CAF phenotype (Fig. 3m). The Gene Ontology analysis implied that Foxf2 may suppress the activity of key transcription factors involved in the immune response. Western blot analysis confirmed that expression of Foxf2 in mouse prostate stromal

cells suppressed lipopolysaccharide (LPS)-induced NF-κB activation and IL-6-induced STAT3 activation (Fig. 5f).

Since both NF-κB and STAT3 can transcriptionally upregulate *Cxcl5*, we reasoned that Foxf2 downregulates *Cxcl5* by repressing activation of either NF-κB or STAT3. Figures 5g, h shows that ectopic expression of the IκB supper-repressor or a dominant negative STAT3$^{Y705S}$ (dnSTAT3) in mPrSCs suppressed *Cxcl5*. However, inhibiting the NF-κB and STAT3 activity did not abolish the downregulation of *Cxcl5* by Foxf2, suggesting that Foxf2 can also regulate *Cxcl5* independent of NF-κB or STAT3. We identified two Foxf2 consensus binding sites in the promoter region of *Cxcl5* (Fig. 5i). ChIP assay showed that Foxf2 directly bound these sites in the mPrSC cells (Fig. 5i). Figure 5j shows that Foxf2 suppressed the luciferase reporter activity driven by a 2 kb *Cxcl5* promoter fragment containing these Foxf2-binding sites. Disrupting the Foxf2-binding sites (Fig. 5j) but not the adjacent NF-κB binding site (Fig. 5k) in the *Cxcl5* promoter ablated the suppression of the reporter activity by Foxf2, demonstrating that Foxf2 can also transcriptionally repress *Cxcl5* independent of NF-κB. Collectively, these results show that Foxf2 can directly transcriptionally suppress *Cxcl5* and indirectly suppress *Cxcl5* by inhibiting the activity of NF-κB and STAT3.

## Stromal Foxf2 sensitizes RM-1 to immune checkpoint inhibitors (ICIs)

Since increased stromal Foxf2 expression enhances T-cell infiltration in several prostate cancer models, we reasoned that increasing stromal Foxf2 activity can enhance the response of prostate cancer to ICIs. As shown in Fig. 6a, we inoculated RM-1 cells with the control and Foxf2-expressing mPrSC cells in C57BL/6 male hosts and treated them with anti-CTLA4, or anti-PD-1, or corresponding control IgGs. Figures 6b, c shows that stromal Foxf2 overexpression attenuated tumor growth as expected. Anti-CTLA4 suppressed the growth of the RM-1 xenografts in the control stromal group by 38% (Fig. 6b). Anti-CTLA4 further suppressed the growth of the RM-1 xenografts in the Foxf2-expressing group. The RM-1 xenografts in the control group were not responsive to anti-PD-1 (Fig. 6c). But anti-PD-1 significantly reduced the weight of the RM-1 xenografts in the Foxf2-expressing group by 48%. Flow cytometric analysis confirms that stromal Foxf2 expression promoted CD8$^+$ T-cell infiltration (Fig. 6d), enhanced T-cell activation (Figs. 6e, f), reduced MDSC, and induced an inflammatory macrophage phenotype (Fig. 6g). Anti-CTLA4 and anti-PD-1 further enhanced these changes. These results demonstrate that increasing stromal Foxf2 activity can mediate an additive tumor suppressive effect of anti-CTLA4 and confers anti-PD-1 responsiveness in the context of the RM-1 tumor model.

We further investigated whether a higher stromal FOXF2 expression in human prostate cancer specimens correlates to a more immunocompetent tumor microenvironment. We laser-captured cells in the stromal regions of prostate cancer specimens of different Gleason patterns and determined the expression of *FOXF2* and genes associated with immune cells and chemokines (Supplementary Fig. 10). We identified a significant positive correlation between *FOXF2* and *CD8a*, although there is no significant correlation between *FOXF2* and *CD3* or *CD4*. There are also significant inverse correlations between *FOXF2* and the M2 macrophage markers *(CD163 and CD206)*, *CXCL6*, *CXCL8*, and *PDCD1*. There is also a trend for an inversed correlation between *FOXF2* and *CXCL5* and *S100A9* although the trend is not statistically significant. These results further support that stromal FOXF2 induces a tumor suppressive TIME.

## Lung stromal Foxf2 suppresses lung metastases of prostate cancer

Foxf2 is expressed by the stromal cells in other endoderm-derived organs such as lung and small intestine. We sought to extend our finding to the lung, where human prostate cancer cells can metastasize

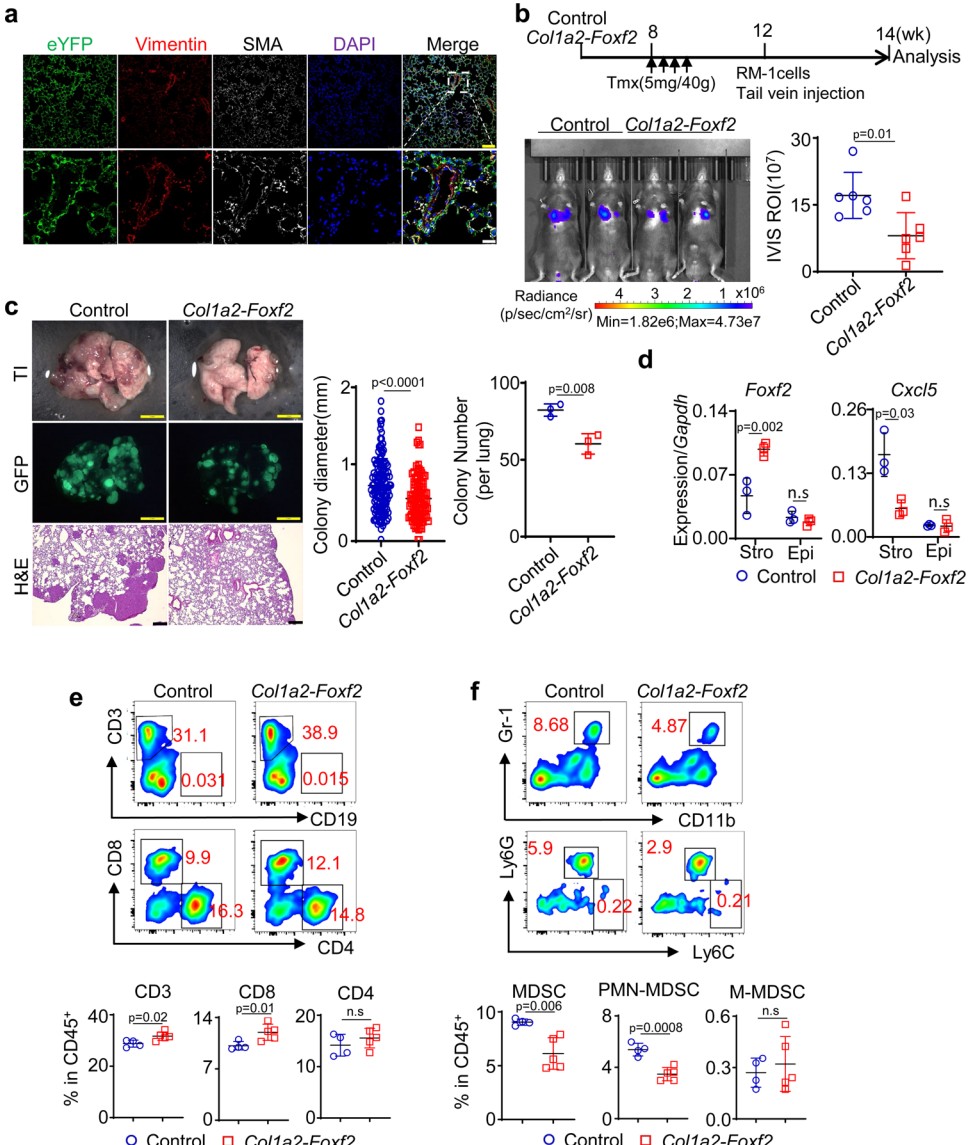

**Fig. 7 | Increasing lung stromal Foxf2 suppresses prostate cancer metastasis.** **a** Representative image of 3 independent Co-immunostaining of eYFP, Vimentin, and smooth muscle actin (SMA) in lungs of 10-wk-old *Col1a2-Foxf2* mice at low (upper panels) and high (lower panels) magnification. Yellow bar = 100 μm; white bar = 25 μm. **b** Schematic illustration of experimental procedure. Image shows bioluminescence imaging of experimental mice. Dot plot shows means ± s.d. of bioluminescent signaling from 6 mice per group. Two-sided unpaired t-test. **c** Transilluminating (upper panels) and fluorescent (middle panels) images, and H&E staining (bottom panels) of lungs from control and *Col1a2-Foxf2* mice received RM-1 cells via tail vein. Dot plots show means ± s.d. of diameter of lung metastases and colony number per lung. In colony diameter plot, each dot represent value of

one metastatic focus. Result is from a total of 247 lung metastatic foci in 6 mice. In colony number plot, each dot represent value from one experimental mouse. Yellow bar = 2 mm; black bar = 250 μm. *N* = 3 per group. Two-sided unpaired t-test. **d** qRT-PCR of *Foxf2* and *Cxcl5* in FACS-isolated lung stromal cells and epithelial cells from lung tissues of tamoxifen-treated *Col1a2-Foxf2* mice. Data represent means ± s.d. of expression from 3 specimens. Two-sided unpaired t-test. **e**, **f** FACS plots of T cells (**e**) and monocytic (M-) and polymorphonuclear (PMN-) myeloid derived suppressor cells (MDSC) in lung metastases. Data represent means ± s.d. of cell percentage from independent tumors from control (*N* = 5) and *Col1a2-Foxf2* mice (*N* = 6) group. Two-sided unpaired t-test. Source data are provided as a Source Data file.

to. Figure 7a confirms that the *Col1a2-CreER^{T2}* model can successfully target the SMA+ or Vimentin+ stromal cells surrounding the lung alveoli as has been reported previously[43]. To test whether increased stromal Foxf2 in the lung can affect lung colonization of metastatic prostate cancer cells, we injected RM-1 cells expressing the firefly luciferase and GFP through tail veins into the tamoxifen-treated *Col1a2-CreER^{T2};R26-LSL-Foxf2^{Tg/Tg}* (*Col1a2-Foxf2*) and *Col1a2-CreER^{T2}* control mice. Bioluminescent imaging shows that the RM-1 cells formed 47% less lung metastasis in the *Col1a2-Foxf2* group than the control group (Fig. 7b). Fluorescent images and H&E staining (Fig. 7c) further confirm that both average colony number and size in the *Col1a2-Foxf2* group were significantly reduced than those in the control group. QRT-PCR

analysis of FACS-isolated Lin-EpCAM- lung stromal cells confirmed that *Foxf2* was upregulated by 2.1-fold while *Cxcl5* was reduced by 66% in the *Col1a2-Foxf2* group (Fig. 7d). We dissociated the metastatic tumor tissues into single cells and analyzed the immune cell composition by flow cytometry. We consistently observed an increase in the CD8+ T cells (Fig. 7e) and a decrease in MDSC (predominantly PMN-MDSC) (Fig. 7f), although the polarity of neither infiltrating nor alveolar macrophage was altered significantly (Supplementary Fig. 11). This result demonstrates that increased expression of stromal Foxf2 in the lung can suppress tumor metastasis, which may account for the reduced distant metastases in the *Cola2-Foxf2-TRAMP* model (Fig. 3f).

## Discussions

Our study shows that a higher stromal Foxf2 expression hampers the generation of an immunosuppressive microenvironment and facilitates antitumor immunity. This finding can potentially explain why the human prostate cancer is relatively rare and indolent in the TZ than PZ prostate since *FOXF2* is expressed at a higher level in the TZ stroma. Unfortunately, we were not able to definitively establish the link because information regarding tumor location is usually not documented in the human prostate cancer datasets. In the TCGA dataset, there are 8 prostate cancer specimens that were clearly documented as the TZ tumors. The average expression level of *FOXF2* was lower in the PZ than in those TZ tumors but the difference did not reach statistical significance (expression ratio in PZ versus TZ = 0.59, $p = 0.15$, t-test). Interestingly, a tumor-associated macrophage (TAM) signature scored higher in the PZ tumors than the TZ tumors ($p = 0.012$, t-test). In addition, *CXCL6*, the human homolog of mouse Cxcl5, was also significantly higher in PZ ($p = 0.012$, t-test). However, *CD8A* was expressed at a higher level in PZ ($p = 0.02$, t-test). Therefore, we cannot conclude whether the tumor-immune microenvironment in PZ is more tumor permissive. Data from more cases are needed to increase the power of these bioinformatic analyses. We propose that the information about the anatomic locations of primary tumors should be included in clinical databases in the future. It is widely accepted that resident tissue fibroblasts can be educated by and coevolved with cancer cells and serve as one source of CAFs. Our study highlights that tissue resident fibroblasts in distinct anatomic locations within the same organ may play differential roles in tumor initiation and progression.

We showed that Foxf2 suppressed the activities of NF-κB and STAT3 in stromal cells. This is consistent with a shift of iCAF to myCAF phenotype observed in the *Col1a2-Foxf2-TRAMP* model because NF-κB and STAT3 were shown to be critical for the development of iCAFs in pancreatic cancer[44]. Therefore, our study establishes Foxf2 as a transcription factor in the stromal cells that modulates tumor immune microenvironment. We show that this change in the tumor immune microenvironment in the prostate and lung can suppress tumor growth and metastasis and sensitize prostate cancer cells to ICIs. Stromal Foxf2 may also mediate tumor suppressive effects via other mechanisms. Stromal Foxf2 has been shown to suppress intestinal adenoma formation through direct stromal-epithelial interaction[15,20]. Although our in vitro and in vivo tumor-stromal cell coculture data did not support a role of this mechanism in the models we tested, it may work in other settings. Additionally, although we showed that stromal Foxf2 did not delay peripheral tolerance towards the SV40 T IV antigen, the tolerance towards other tumor antigens might be affected. Based on these observations and arguments, increasing Foxf2 expression alone or in combination with immunotherapies could be a therapeutic strategy for prostate cancer. Although directly manipulating FOXF2 is difficult, it is possible to identify secreted factors or small molecule compounds that can upregulate stromal *FOXF2* expression, which can be therapeutically exploited. We also showed that increasing Foxf2 expression in lung stroma can suppress prostate cancer colonization. It is tempting to speculate that in those tissues with endogenous stromal Foxf2 expression such as lung and small intestine[14], increasing Foxf2 expression might also attenuate immunosuppressive tissue microenvironment. Finally, although we demonstrated that Cxcl5 is a major downstream effector of stromal Foxf2 in the RM-1 model, we cannot exclude potential roles of other cytokines in human cancers. For example, we showed that the expression of *CXCL6* and *CXCL8*, the functional homologs of mouse *Cxcl5*, are both inversely correlated with that of *FOXF2* in human prostate cancer specimens.

Foxf2 is expressed in the stem or progenitor cell-like cells such as pericytes, progenitors of smooth muscle cells, or mesenchymal stem cells[18,45]. It is unclear whether the differential expression level of Foxf2 in TZ and PZ stroma reflects a different developmental state of the stromal cells. The heterogeneity of mouse and human prostate stromal cells has been revealed through immunostaining, single-cell analysis, and lineage tracing studies[36,46]. Peng et al. defined a few stromal populations in mouse prostate: subepithelial fibroblasts adjacent to the basement membrane, smooth muscle cells, and the interstitial fibroblasts in between prostatic ducts[46]. Our RNA-In-Situ analysis shows that Foxf2 is highly expressed by the subepithelial fibroblasts in mouse prostates (Fig. 1d). Foxf2 is regulated by the Shh signaling pathway[47]. Prostate basal cells express the Shh ligands, whereas the subepithelial fibroblasts express Gli1[46]. Therefore, the expression pattern of Foxf2 is likely shaped by epithelial-stromal interaction. These cells are very much like the Gli1+ stromal cells in the pancreas[48]. In mouse prostate, the density of basal cells is higher in the proximal prostate, which may explain why Foxf2 is highly expressed by the stromal cells at proximal ducts than in those of distal ducts. In the human prostate, the expression of *FOXF2* is not restricted to the subepithelial stromal cells (Fig. 1a). Nevertheless, TZ stroma express *FOXF2* more than PZ stroma. It is possible that the TZ basal cells express shh at a higher level or other signaling also contribute to the distinctive expression pattern of *FOXF2*.

The differential expression pattern of Foxf2 plays a role in tissue morphogenesis. We showed previously that the proliferation of epithelial cells in the mouse proximal prostate proliferate less because branching morphogenesis in this region should be restricted to avoid overcrowded glands. The low proliferative index is maintained by a low epithelial cell-intrinsic canonical Wnt signaling and a high expression of Wnt5a, Sfrps, and Tgfβ in adjacent stromal cells[23]. Stromal Foxf2 can regulate Sfrp1 and Tgfβ[15,18]. Therefore, differential expression of Foxf2 may also play a critical role in maintaining normal tissue homeostasis, although our study excludes such regulation in the context of the tested models. Dysregulation of its expression may contribute to the development of prostate-related diseases including both prostate cancer and benign prostatic hyperplasia (BPH). A hallmark of prostate cancer is the loss of the prostate basal cells. It is tempting to speculate that loss of basal cells downregulates the expression of Foxf2, which induces tumor permissive immune microenvironment and assist in prostate cancer progression. In BPH, increased shh signaling may upregulate Foxf2 and regulate Tgfβ.

Although many reports support that prostate cancer in TZ is relatively less aggressive, others suggest otherwise. For example, African American men with very low risk diseases at biopsy are at a greater risk for adverse oncologic outcomes than Caucasian patients[49]. This could be because African American patients tend to have high-grade dominant nodules at the anterior aspect of the prostate gland where TZ sits[50,51]. These observations can be potentially explained by our previous study showing that the properties of the cells of origin for prostate cancer can determine its clinical features[24,52]. Specifically, tumors originating from the cells in the mouse proximal prostatic ducts and human TZ zone were predisposed to castration resistance and lineage plasticity. Therefore, the clinical behavior of prostate cancer could be determined by both the property of its cell-of-origin and the tissue microenvironment.

## Methods

All animals used in this study received humane care in compliance with the principles stated in the Guide for the Care and Use of Laboratory Animals, NIH Publication, 1996 edition, and the protocol was approved by the Institutional Animal Care Committees of Baylor College of Medicine and University of Washington. In all the studies, the maximal allowable tumor size/burden (diameter less than 1.5 cm) was not exceeded. All human specimens were collected with informed consent according to the guidelines of the Declaration of Helsinki and approved by the Institutional Review Board (or Ethics Committee) of University of Washington (protocol code 2341).

### Mice

Mice were housed in AAALAC-accredited vivarium. Room temperature was maintained at $22 \pm 3\,°C$ and the humidity at $55 \pm 15\%$. Animals were kept at a 12 h/12 h dark/light cycle. The C57BL/6 and SCID/Beige mice were purchased from Charles River (Wilmington, MA). Nude mice, *Col1a2-CreER$^{T2}$*, *C57BL/6-Tg(TRAMP)8247Ng/J*, and *B6.Cg-Gt(ROSA) 26Sor$^{tm3(CAG-EYFP)Hze}$/J*, *Pten$^{tm1Hwu}$*, and *R26-LSL-Kras$^{G12D}$* mice were purchased from the Jackson Laboratory (Bar Harbor, ME). The *ARR2Pb-Cre* mice were from Dr. Fen Wang at the Institute of Biosciences and Technology at Texas A&M. The *R26-LSL-Foxf2* mice were generated at the Baylor College of Medicine. The *Foxf2* null mouse embryos were generously provided by Dr. Rulang Jia at Cincinnati Children's Hospital. All mice were on the C57BL/6 background. Mice were genotyped by polymerase chain reaction using mouse genomic DNA from tail biopsy specimens. The sequences of genotyping primers and the expected band sizes for PCR are listed in Supplementary Table 2. PCR products were separated electophoretically on 1% agarose gels and visualized via ethidium bromide under UV light.

### Human specimens

The human PZ and TZ prostate specimens and sections of human prostate cancer specimens were obtained from patients underwent radical prostatectomy for the treatment of prostate cancer at the University of Washington at Seattle. Collected specimens were fixed in 10% formalin at $4\,°C$ for 12 h. Tissue sections were reviewed by pathologists to confirm that the areas of benign specimens used in the study were devoid of tumor or benign prostatic hyperplasia. Immunostaining using antibodies against CK5 and Trp63 was also performed to confirm the absence of tumor.

### Tamoxifen and BrdU treatment

Tamoxifen (Sigma-Aldrich, St. Louis, MO) was dissolved in corn oil and injected i.p. into experimental mice at the specified age (5 mg/40 g/day for 4 consecutive days). BrdU (Sigma-Aldrich, St. Louis, MO) (80 mg/kg/day) was administrated 12 h before mice were sacrificed.

### Tumor xenograft and bioluminescence imaging

For subcutaneous tumor xenografts, $1.5 \times 10^5$ mouse prostate stromal cells were mixed with $5 \times 10^4$ RM-1 or Pten/Kras cells in 100 µl PBS and injected subcutaneously into 8 to 10-week-old male SCID/Beige, Nude, or C57BL/6 hosts. Experiments were terminated as planned or once the diameter of tumor exceeded 1.5 cm. For the study of lung metastasis, 10-week-old male *Col1a2-CreER$^{T2}$;LSL-Foxf2* and age-matched littermate control *Col1a2-CreER$^{T2}$* mice were treated with tamoxifen (5 mg/40 g/day for 4 consecutive days). 21 days after the treatment, $2 \times 10^5$ RM-1 cells in 100 µl PBS were injected into mice through tail veins. 14 days later, the mice were imaged using an IVIS Lumina II (Advanced Molecular Vision) following the manufacturer's recommended procedures and settings.

### Cell culture

Single cells were prepared from 8 to 12-week-old C57BL/6 mouse prostate tissues as described previously[37]. Dissociated single cells were cultured in Biocoat Collage I-coated plates (Corning, Corning, NY) in Bfs medium (5% Nu-Serum, 5% FBS, 1 × Insulin/Selenium, 1 × L-Glutamine, 1 × Penicillin/Streptomycin, and $1 \times 10^{-10}$ M DHT in DMEM medium) at $37\,°C$ with 5% $CO_2$. Cells were trypsinized into single cells with 0.25% Trypsin-EDTA (Invitrogen, Carlsbad, CA) when reaching 90% confluency. Cells were replated in Biocoat™ Collage I-coated plates (Corning, Corning, NY) for 30 min at $37\,°C/5\%$ $CO_2$. Unattached cells were discarded, and remaining cells were cultured in Bfs at $37\,°C/5\%$ $CO_2$ till 80–90% confluency. All the experiments in this study used fresh primary stromal cells within 3 passages after single-cell dissociation from prostates. 6-10 $\times 10^5$ stromal cells can be obtained after a 3-passage culture of stromal cells from one 8-week-old mouse prostate.

RM-1 cells were a kind gift from Dr. Timothy Thompson at the University of Texas MD Anderson Cancer Center and were cultured in 10% FBS in DMEM at $37\,°C/5\%$ $CO_2$. WPMY-1 cells were a kind gift from Dr. Chawnshang Chang at the University of Rochester Medical Center and cultured in 10% FBS in DMEM at $37\,°C/5\%$ $CO_2$. The *Pten-Kras* cell line was established from the primary prostate tumor tissues of *Pb-Cre;Pten$^{fl/fl}$;LSL-Kras$^{G12D}$* mice. Briefly, tumors were cut into small pieces of approximately 1 mm$^3$. Tissues were digested in DMEM/Collagenase/Hyaluronidase/FBS (Invitrogen, Carlsbad, CA) for 1 h at $37\,°C$, followed by an additional 1 h of digestion in 0.25% Trypsin-EDTA (Invitrogen, Carlsbad, CA) on ice. Cells were then passed through 70 µm cell strainers (BD Biosciences, San Jose, CA) to get single cells. Dissociated single cells were cultured in DMEM with 10% FBS, 6 ng/mL EGF (Invitrogen, Carlsbad, CA), and 5 µg/mL insulin (Invitrogen, Carlsbad, CA). Tumor cells were separated further from the Sca-1-expressing stromal cells by flow cytometry. Cells were maintained at $37\,°C/5\%$ $CO_2$.

### Dissociation of mouse prostate tissues and RM-1 xenografts

Prostate tissues were dissociated into single cells according to the procedure described previously[53]. Briefly, mouse prostate tissues or RM-1 tumors were incubated in DMEM/Collagenase/Hyaluronidase/FBS (STEMCELL technologies, Vancouver, Canada) for 3 h at $37\,°C$, followed by one hour-incubation in 0.25% Trypsin-EDTA (Invitrogen, Carlsbad, CA) on ice. Thereafter, mouse prostate tissues were pelleted, resuspended in Dispase (Invitrogen, Carlsbad, CA, 5 mg/mL) and DNase I (Roche Applied Science, Indianapolis, IN, 1 mg/mL), and pipetted vigorously to dissociate cell clumps. Dissociated cells were then passed through 70 µm cell strainers (BD Biosciences, San Jose, CA) to obtain single cells.

### Preparation of RM-1 cell lysates for cytokine measurement

100 mg tumor tissue was added in 500 µL precooled extraction buffer [10 mM Tris–HCl (pH 8.0), 150 mM NaCl, 1% NP-40, 10% Glycerol, 5 mM EDTA, Protease inhibitor cocktail (Roche), Phosphatase inhibitor cocktail, 1 mM PMSF (Phenyl Methyl Sulfonyl Fluoride)] in an Eppendorf tube and homogenized using Bel-Art ProCulture Micro-Tube Homogenizer. The homogenate was sonicated using a Virtis Virsonic 100 Ultrasonic Cell Disrupter (SP Industries, Warminster, PA) and incubated on ice for 20 min and centrifuged at $14,000 \times g$ for 15 min at $4\,°C$. The supernatant was collected, and the protein concentration was measured by the Bradford protein assay (BioRad, Hercules, CA), aliquoted, and stored at $-80\,°C$.

### Flow cytometry and cell sorting

Dissociated prostate or tumor cells were incubated with florescence conjugated antibodies at $4\,°C$ for 30 min. Information for antibodies for FACS analyses and sorting is listed in Supplementary Table 3. FACS analyses and sorting were performed using BD LSR II, BD LSR Fortessa, and Aria III (BD Biosciences, San Jose, CA) and analyzed using BD FACSDiva software and Flowjo (Flowjo LLC, Ashland, OR).

### RNA isolation and quantitative RT-PCR

Total RNA was extracted using Nucleospin RNA extraction Kit (Macherey-Nagel, Bethlehem, PA), reverse transcribed to cDNA using iScript Reverse Transcriptase Kit (BioRad, Hercules, CA). QRT-PCR was performed using iTaq Universal SYBR Green Supermix (BioRad, Hercules, CA) and detected on a QuantStudio5 Real-Time PCR system (Applied Biosystems, Foster City, CA). Primers for target genes were listed in Supplementary Tables 4 and 5.

### RNA-Scope

Mouse prostate tissues or human TZ and PZ tissues collected from the same donors were fixed in 10% neutral buffered formalin for 16-32 h at room temperature and embedded in paraffin. 5-micron sections were cut, air dried overnight at room temperature and baked for 1 h at $60\,°C$.

The RNA-Scope in situ hybridization was performed by using RNA-Scope 2.5 HD Red or Duplex Detection Reagent Kit (Advanced Cell Diagnostics, Newark, CA) following the manufacturer's standard protocol. 20–60 images were taken for each sample to cover all areas of stained specimens. Results were analyzed using the Image-Pro Plus version 6.3 by Media Cybernetics. Nuclei numbers and areas with staining of *FOXF2* in the defined areas were determined by the count feature in the software. Total *FOXF2* staining areas within the stromal cells were normalized by nucleus number of stromal cells on each image.

## Visium spatial gene expression assay and data analysis

Frozen sections of urethra and connecting prostate tissues from C57BL/6 mice were fixed for H&E staining and imaging following the manufacturer's instruction (CG000160, 10X Genomics, Pleasanton, CA). Tissues were permeabilized for 12 min and spatially barcoded full-length cDNA was generated using Visium Spatial Gene Expression Slide & Reagent Kit (PN-1000187, 10X Genomics, Pleasanton, CA). CDNA was amplified by 14 cycles of PCR reaction before library preparation. The libraries were sequenced with NextSeq 1000/2000 P2 Reagents 100 cycles (Cat#20043738, Illumina, San Diego, CA) on a NextSeq2000 (Illumina), with Read 1: 28 cycles, i7 index:10 cycles, i5 index: 10 cycles, and Read 2: 90 cycles.

Raw sequencing data were converted to FASTQ format using BCL Convert v1.2.1 on Illumina's BaseSpace sequencing hub. Scanpy (ver.1.8.1), Pandas (1.1.5), Scipy (1.4.1), Seaborn (0.11.2), and Matplotlib (3.2.2) were used for downstream data processing, visualization, and statistics. Raw spatial transcriptomics data were normalized using Scanpy preprocessing APIs. Gene counts per cell were normalized and logarithmized. Spatial profiles of distal and proximal prostatic ducts were defined by aligning both histological features and spatial gene clusters identified using Scanpy clustering APIs. For Leiden cluster analysis, the dimensionality of the expression data was reduced to 50 with principal component analysis and then computed for neighborhood connectivity with a local neighborhood size of 15. Leiden clustering algorithm from Scanpy with a resolution of 0.1 consistently revealed spatial cluster distributions that closely resemble histological features of distal and proximal prostatic ducts across all the samples. "Contaminating" Visium spots from the clusters based on their histology were excluded from downstream analysis. Individual clusters were labeled based on their histological identities for further gene expression analysis. The expression levels of *Foxf2* at distal and proximal prostatic ducts were visualized using the violin plot API of the Seaborn package. The Scipy package was used for statistical tests and *p*-values.

## Lentivirus preparation

Oligos were ordered from Integrated DNA Technologies, Inc. (Coralville, Iowa). The shRNAs include the scrambled shRNA (5'CCT AAGGTTAAGTCGCCCTCGCTCGAGCGAGGGCGACTTAACCTTAGG3'), the *Cxcl5* shRNAs (5'CCGGGAAGTCATAGCTAAACTGAAACTCGAGTTT CAGTTTAGCTATGACTTCTTTTTG3' and 5'CCGGCGGAATGCACTC GCAGTGGAACTCGAGTTCCACTGCGAGTGCATTCCGTTTTTG3'), the *Cxcl9* shRNAs (5'CCGGGTCGTCGTTCAAGGAAGACTACTCGAGTAG TCTTCCTTGAACGACGACTTTTTG3' and 5'CCGGCATCATCTTCCTG-GAGCAGTGCTCGAGCACTGCTCCAGGAAGATGATGTTTTTG3'), the *Cxcl10* shRNAs (5'CCGGTCCGGAATCTAAGACCATCAACTCGAGTTG ATGGTCTTAGATTCCGGATTTTTG3' and 5'CCGGTCCGGAATCTAA GACCATCAACTCGAGTTGATGGTCTTAGATTCCGGATTTTTG3'), the *Foxf2* shRNA

(5'CCGGCCAAAGAACATTGTGAAGAAACTCGAGTTTCTTCACAAT GTTCTTTGGTTTTTG 3'). The U6-shRNA-expressing cassettes and the cDNAs for *Foxf2*, super *IκB*, and dominant negative *STAT3* were cloned into the FU-CGW or FU-CRW lentiviral vector[54] at the *PacI*, and *EcoRI* sites, respectively. Lentivirus preparation, titering, and infection of

mouse prostate stromal cells or WPMY-1 cells were performed as described previously[54].

## RNA-seq

RNA was extracted from control and Foxf2-expressing mouse prostate stromal cells and WPMY-1 cells using MasterPure RNA Purification Kit (Illumina, San Diego, CA). TruSeq Stranded mRNA Sample Preparation Kit (Illumina, San Diego, CA) was used to prepare cDNA libraries, which were sequenced using HiSeq 2500 sequencer. Sequenced reads in FASTQ files were mapped to mm10 whole genome using Tophat2, and Fragments Per Kilobase of transcript per Million mapped reads (FPKM) were calculated using Cufflinks. Genes found differentially expressed (*p* < 0.05 by t-test, and minimum fold change of 1.4 or 1.2) were evaluated for enrichment of Gene Ontology (GO) gene classes, using SigTerms software[55]. Data have been deposited at GEO.

For CAF gene signature scoring, log2-transformed expression values in the Foxf2 RNA-seq dataset were first centered on the sample median, and then the sum of centered values in the Liu et al. signature[41] for each sample was computed. For the Mishra et al. signature[42], the sum of the "DOWN" gene values was subtracted from the sum of the "UP" gene values for each sample. The signature scores were then normalized across samples to standard deviations from the median.

## Single-cell RNA-sequencing (scRNA-seq)

Sorted cells were washed and resuspended in PBS containing 1% BSA. Cells were counted on Countess II automated cell counter (Thermo Fisher), and up to 10,000 cells were loaded per lane on 10X Chromium microfluidic chips. Single-cell capture, barcoding, and library preparation were performed using the 10X Chromium Next GEM Single Cell 3' GEM, Library & Gel Bead Kit v3.1 (PN-1000128, 10X Genomics) following the manufacturer's protocol. cDNA amplification was conducted with 12 cycles and 10 μl cDNA was used in library preparation protocol of fragmentation, adapter ligation, and indexing. cDNA and libraries were checked for quality on an Agilent 4200 TapeStation using high sensitivity D5000 ScreenTape (Cat#5067-5592, Agilent Technologies, Santa Clara, CA) and reagents (Cat#5067-5593, Agilent) and the libraries (650 pM with a final volume of 20 μL) were sequenced with NextSeq 1000/2000 P3 Reagents 100 cycles (Cat#20038730, Illumina, San Diego, CA) on a NextSeq2000 (Illumina), with 28 cycles for read 1, 8 cycles index 1, and 91cycles for read 2.

The raw sequencing data was converted to FASTQ format using BCL Convert v1.2.1 on Illumina's BaseSpace sequencing hub. Sequencing data then were uploaded to the 10x Genomics Cloud Analysis platform (www.10xgenomics.com/products/cloud-analysis, 10x Genomics) and processed with the Cell Ranger Count v6.1.1 pipeline using the Mouse (mm10) 2020-A reference genome and default settings. The resultant output files were used as inputs for analysis in Seurat v4.0.5 in R v4.1.0 (www.R-project.org). Briefly, the immune and stromal datasets were handled separately with cells filtered by number of genes and percent mitochondrial genes, scaled and clustered, subjected to UMAP dimension reduction, and filtered to remove contaminating cell types based on cell type specific gene expression. For the immune dataset clusters highly enriched for *Epcam* and *Krt18* (epithelial), and *Acta2* and *Dcn* (stromal) markers were removed. For the stromal dataset, one cluster highly enriched for *Epcam* and *Krt18* (epithelial) was removed. Cell type- or subtype-specific marker genes were used to identify cell types represented by clusters/subclusters for immune cells: Macrophage – *Adgre1*, T-cell – *Cd3g*, CD8+ T-cell – *Cd8b1, Cd8a*, CD4+ T-cell – *Cd4*, Proliferative T-cell – *Mki67*, Plasma cell – *Jchain*, B-cell – *Ms4a1*, NK cell – *Prf1*, Monocyte– *Sell*, pDC – *Siglech*, cDC – *Xcr1*. For stromal cell population cluster annotation was performed manually using the following marker genes: myCAF – *Acta2*, iCAF – *Dcn, Pdgfra*, Proliferative – *Mki67*, Perivascular – *Rgs5*. Cell type annotations were appended to the metadata as was sample ID (control or Foxf2). Differential expression and corresponding statistics (logfc

and adjusted *p* value) were computed using the FindAllMarkers command with the minimum percent of cell expressing, "min.percent," parameter set to 0.01 and the logfc threshold set to 0.1. Violin plots, gene expression heatmaps, and UMAP plots were generated using the corresponding Seurat commands. For the stromal dataset, a CAF signature score was calculated using the following 30 genes in Seurat's AddModuleScore function: *S100a4, Acta2, Zeb1, Slc16a4, Pdpn, Foxf1, Fap, Vim, Pdgfrb, Sparc, Pdgfra, Mmp2, Mmp11, Aspn, Fn1, Mfap5, Ogn, Tnc, Col3a1, Col11a1, Col1a1, Col1a2, Emilin1, Col5a1, Col16a1, Loxl1, Thy1, Ly6a, Il-6, Has1*, and with 100 control genes randomly selected from the same bin per analyzed gene. CAF signature scores were compared for control versus Foxf2 samples using a two-sided Wilcoxon Rank Sum test where individual cells were taken as samples. This analysis was performed using the wilcox.test function in the R package, "stats".

## ChIP assay

Primary mouse prostate stromal cells were crosslinked in cell culture media containing 1% methanol free formaldehyde (Polysciences, Warrington, PA) for 10 min at room temperature. Cross-linking was terminated with 125 mM Glycine for 5 min at room temperature. Cells were harvested in PBS containing protease inhibitors and PMSF, resuspended in sonication buffer and sonicated using a bioruptor (Diagenode Inc., Denville, NJ). Input samples were taken following sonication. The remaining sonication product was divided into 2 halves and incubated with 4 µg antibodies against FLAG tag (F1804, Sigma-Aldrich, St. Louis, MO) or normal rabbit IgG (sc-2027, Santa Cruz Biotech, Santa Cruz, CA) in ChIP Buffer 1 containing protease inhibitors and magnetic G beads overnight at 4 °C in siliconized tubes. Beads were collected and washed with ChIP Buffer 1 and ChIP Buffer 2 and then resuspended in elution buffer for 15 min at room temperature. Formaldehyde crosslinks were reversed using reverse cross-linking buffer at 65 °C for 2.5 h. Remaining proteins were digested with Proteinase K (50 µg/µl) at 37 °C for 1 h. DNA was isolated using the Purelink Quick Gel Extraction Kit (Invitrogen, Carlsbad, CA). qRT-PCR was performed to determine enrichment using the primers listed in Supplementary Table 6. Ct values from α-IgG and α-FLAG were normalized against Ct values generated from the input samples. The resulting value was then normalized by dilution and concentration to determine the value relative to the input.

## Luciferase reporter assay

A 2 Kb genomic sequence upstream of the transcription start site of mouse *Cxcl5* containing 2 putative Foxf2-binding sites and 1 NF-κB binding site was PCR amplified from a BAC clone from the BACPAC Resources Center (Children's Hospital Oakland Research Institute) containing the corresponding genomic region using LA Taq (Takara Bio Inc., Otsu, Shiga, Japan). The primers used are listed in Supplementary Table 7. The amplicon was cloned into the pGL3 luciferase vector (Promega, Madison, WI) via the *KpnI* and *BglII* restriction sites upstream of *luciferase*, generating the pGL3-Cxcl5-luc reporter. Mutations at the Foxf2 and NF-κB binding sites were performed by site-directed-mutagenesis using the Q5 Site-Directed Mutagenesis kit (New England Lab, Woburn, MA).

Primary mouse prostate stromal cells were infected with either FU-CRW or FU-Foxf2-CRW. Two days later, cells were seeded in 6 well plates and co-transfected with 40 ng of pRL-CMV Renilla and 2 µg of pGL3-Cxcl5-luc or corresponding mutant constructs, respectively, using lipofectamine 3000 according to the manufacturer's instructions. Two days after transfection, the Bfs medium was replaced with 1:10 Opti-MED diluted Bfs medium. 24 h later, luciferase activity was measured using the dual-Luciferase reporter assay system (Promega, Madison, WI). Firefly luciferase activity was normalized to CMV-Renilla luciferase activity. Data were presented relative to CMV-Renilla

readings and shown as mean ± s.d. Experiments were performed in triplicates.

## Western blots and ELISA assay

Cells or RM-1 tumor tissues were lysed in RIPA buffer (20 mM Tris–HCl, pH 7.5, 150 mM NaCl, 1 mM Na$_2$EDTA, 1 mM EGTA, 1% NP-40, 1% sodium deoxycholate, 2.5 mM sodium pyrophosphate, 1 mM β-glycerophosphate, 1 mM Na$_3$VO$_4$) with protease inhibitors and phosphatase inhibitors (Roche Applied Science, Indianapolis, IN). Tissuelyser LT (Qiagen, Valencia, CA) was used to facilitate lysis of tumor tissues. Protein concentrations were determined by a Bradford Assay kit (BioRad, Hercules, CA). Protein was separated by 10% SDS/PAGE and transferred onto a nitrocellulose membrane (Amersham Biosciences, Arlington Heights, IL). The membrane was blocked in 5% skim milk, and subsequently incubated with primary antibodies listed in Supplementary Table 8 at 4 °C overnight followed by incubation with secondary antibodies listed in Supplementary Table 8 and developed with Pierce ECL reagent (Thermal Scientific, Rockford, IL). Mouse Cxcl5 Quantikine ELISA kit (MX000, R&D systems, Minneapolis, MN) was used to determine Cxcl5 level in tumor lysates or cell culture media according to the manufacturer's instruction. The cytokine arrays containing 22 cytokines and chemokines were performed through paid service at Eve Technologies (Calgary, Alberta, Canada).

## Dendritic cells preparation, vaccination, and cytotoxicity assay

Dendritic cells (DCs) were prepared from C57BL/6 bone marrow. Femurs and tibias were removed, and bone marrow was flushed with a 25-gauge needle. After RBC lysis (Invitrogen, Carlsbad, CA), bone marrow cells were plated in 10 cm dishes in complete RPMI 1640 supplemented with 20 ng/mL GM-CSF. On day 8, DCs were matured by culturing with 10 ng/mL of IL4, 20 ng/mL of GM-CSF, and 1 µg/mL LPS for 24 h. Matured DCs were pulsed 1 h with 2 µg/mL of TAG-IV peptide (GenScript, Piscataway, NJ) and injected *i.d.* into the right flank of mice ($5 \times 10^5$ DCs in 50ul PBS/mouse). Mice were sacrificed 7 days later. Cell suspensions were obtained from mechanical disaggregation of spleen. For intracellular detection of IFNγ, splenocytes were stimulated 4 h with TAG-IV peptide (1 µg/mL). Brefeldin A (10 µg/mL) was added in the last 3 h. Cells were stained for surface markers, fixed with IC fixation buffer (Invitrogen, Carlsbad, CA), permeabilized with permeabilization buffer (Invitrogen, Carlsbad, CA), and incubated with anti-IFNγ. Samples were acquired with BD LSR II Fortessa and analyzed with the FlowJo software.

## qRT-PCR from Laser captured Frozen tissues

Slides of 6 µm were sectioned from Frozen blocks using a Leica CM 1950 microtome, mounted onto Arcturus PEN Membrane Frame Slides (Thermo Scientific, Waltham, MA), fixed with 95% ethanol, and stained with Cresyl Violet (Acros Organic, New Jersey, NJ). Adjacent sections of 5 µm were cut and stained with hematoxylin and eosin. Histology review and slide annotation was performed by a pathologist. Areas of benign and Gleason patterns were marked if present. Areas of stroma and tumors were captured using the Arcturus XT (Thermo Scientific, Waltham, MA) with CapSure Macro LCM Caps (Thermo Scientific, LCM0211) and RNA was extracted using the Arcturus PicoPure RNA Isolation kit (Thermo Scientific, Waltham, MA) according to manufacturer's recommendations. RNA quality (DV200) and quantity were assessed using the TapeStation 4200 (Agilent Technologies, Santa Clara, CA) with High Sensitivity RNA Screentape (Agilent Technologies, Santa Clara, CA). RNA was reverse transcribed to cDNA using iScript™ Reverse Transcriptase kit (BioRad, Hercules, CA). cDNA was pre-amplified using SsoAdvanced™ PreAmp Supermix (BioRad, Hercules, CA). QRT-PCR was performed using iTaq Universal SYBR Green Supermix (BioRad, Hercules, CA) and detected on a Quantstudio Real-Time PCR system (Applied Biosystems, Foster City, CA).

## Migration and invasion assays

The migration and invasion assays were carried out using 8.0 μm pore-size transwell chambers (Corning, Tewksbury, MA). $1 \times 10^5$ primary stromal cells were cultured at the lower compartment. For the migration assay, $5 \times 10^3$ RM-1 cells resuspended in serum-free DMEM medium were added to the upper compartment of the chamber. For the invasion assay, $1 \times 10^4$ RM-1 cells resuspended in serum-free DMEM medium were seeded in chambers coated with Matrigel (Corning, Tewksbury, MA). After a 24-h-coculture, the migrated and invaded cells on the membranes were stained with 1% crystal violet (Sigma-Aldrich, St. Louis, MO) followed by colorimetric quantification. Briefly, 750 μl 10% acetic acid (Sigma-Aldrich, St. Louis, MO) was filled in the wells to lyse the cells on the membrane and release Crystal Violet. Optical absorbance at 595 nm was measured using a microplate reader (Perkin Elmer, Waltham, MA).

## Histology and immunostaining

Prostate tissues were fixed in 10% buffered formalin and paraffin embedded. H&E staining and immunofluorescence staining were performed with 5 μm sections. For hematoxylin and eosin staining and immunostaining, sections were processed as described previously[56]. For immunostaining, sections were processed as described previously[56] and incubated with primary antibody in 3% of normal goat serum (Vector Laboratories, Burlingame, CA) overnight. Information for primary antibodies is listed in Supplementary Table 8. Slides then were incubated with secondary antibodies (diluted 1:250 in PBST) labeled with Alexa Fluor 488 and 594 (Invitrogen/Molecular Probes, Eugene, OR). Sections were counterstained with either hematoxylin or 4,6-diamidino-2-phenylindole (DAPI) (Sigma-Aldrich, St. Louis, MO).

Immunofluorescence staining was imaged using a Leica SP8 fluorescence microscope (Leica Microsystems, Wetzlar, Germany). Images of IHC were processed using an image-processing package, Fiji (https://fiji.sc/). To quantify nuclear fluorescence intensity of FOXF2, at least 10 images of different fields were taken for each Gleason pattern in individual patient specimens. Images were converted into an 8-bit format. DAPI staining was used to enumerate cell numbers and define stromal region using a polygon tool. Stromal regions containing at least 30 nuclei in benign tissues and in cancerous areas of different Gleason patterns were randomly chosen to determine Mean Fluorescence Intensity (MFI) of FOXF2. MFI was corrected with background fluorescent signaling in peri-nuclear areas as described previously[57]. Data were analyzed by two-way ANOVA to identify significant differences between groups and significance was determined as $p \leq 0.05$. For quantitative analysis of angiogenesis in tumors, CD31 staining was quantified using ImageJ with the VesselJ plug-in (https://imagej.nih.gov/ij/plugins/vesselj/index.html)[58]. Significant difference was determined as $p < 0.05$ by Student's t test.

## Statistical analyses

All experiments were performed using 3–18 mice in independent experiments. Data are presented as means ± s.d. Student's t test and one-way or two-way ANOVA with multiple comparisons were used to determine significance in two-group and multiple-group experiments, respectively. Correlation analysis between expressions of *FOXF2* and other genes was made by Spearman's rank. Frequencies of metastases and neuroendocrine cancers were compared by Fischer's exact test. For all statistical tests, the two-tail $p \leq 0.05$ level of confidence was accepted for statistical significance. More details can be found in the methods or figure legends.

## Data availability

The accession numbers for the RNA-seq data of the WPMY-1 cells, scRNA-seq data for stromal and immune cells in *Col1a2-Foxf2-TRAMP* mice, and Visium analysis of mouse prostate in this paper are GEO: GSE85094, GSE193280 and GSE216219, respectively. Source data are provided with this paper. The remaining data are available within the Article, Supplementary Information or Source Data file. Source data are provided with this paper.

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

## Acknowledgements

We thank the patients and their families for supporting this research, Brenda Nghiem and Lori Kollath for sample identification and processing, and Drs. Jingyue Xu and Rulang Jia at Cincinnati Children's Hospital Medical Center for providing materials and advice regarding Foxf2 immunostaining. This work is supported by R01CA190378 and R01DK107436 (L.X.), the Pritt Family Endowment, P50CA97186, P01CA163227, and the Institute for Prostate Cancer Research (IPCR).

## Author contributions

Conceptualization, L.X. and D.J.; Investigation, D.J., Z.Z., O.K., L.Z., X.W., and M.Y.; Formal analysis: D.J., M.Y., L.X., Y.Z., M.P.R., M.C.R., and C.J.C.; Writing-original draft, L.X.; Writing-Review & Editing, L.X. with input from D.J., M.P.R., M.C.R., and C.J.C; Resources, R.D., P.S.N., M.H., C.M., L.T., and D.W.L.; Funding acquisition, L.X.; Supervision, L.X.

## Competing interests

The authors declare no competing interests.
