## [Peer Review File · Nature Communications]

Stromal FOXF2 suppresses prostate cancer progression and metastasis by enhancing antitumor immunityREVIEWER COMMENTS

Reviewer #1 (Remarks to the Author): with expertise in cancer immunology, prostate cancer

This paper would offer mechanistic explanation to a previous observation showing low incidence of prostate cancer (PC) in the transitional zone because of high expression of FOXF2 (Prostate 69, 1538–1547; 2009). This interpretation was corroborated by other showing that FOXF2 is a target gene for miR-182-5p and that miR-182-5p knockout replace FOXF2 expression toward inhibition of proliferation, migration and invasion of PC cells (PLoS ONE 8, e55502; 2013).

The novelty here passes through the demonstration that FOXF2, produced by stromal fibroblasts, regulates the immune microenvironment determining activation of CD8 T lymphocytes and reduction of MDSC and M2 TAM.

The approach and results are generally sound. The elective in vivo transgenic TRAMP model selected to consolidate conclusion is commended but perhaps not used at the best of its potential, with few debatable points. The statement: “On the C57BL/6 background, it (TRAMP) develops prostatic intraepithelial neoplasia (PIN) at 3 months and prostate cancer with focal neuroendocrine differentiation (particularly in ventral prostate) at 6 months” is to my knowledge incorrect. Differently from the FVB background, C57BL/6 TRAMP mice spontaneous primary neuroendocrine tumor occurs in less than 10% of mice in the ventral lobe, apparently from transformation of the basal layer. This brings up also the potential role of FOXF2 in regulation of basal and luminal specification.

Neuroendocrine trans-differentiation from adenocarcinoma occurs roughly in 30% of TRAMP mice while a similar fraction show adenocarcinoma regression, upon castration. Does FOXF2 protect from NE primary transformation and/or NE trans-differentiation of existing adenocarcinoma or it has a distinct effect on the adenocarcinoma?

Another feature of TRAMP mice is the notable expression of SV40, which is an effective endogenous tumor antigen, recognized by CD8 T lymphocytes and tolerized along tumor progression (Eur J Immunol. 35:66-75; 2005). In a paper aiming at demonstrating that the effect of FOXF2 pass through CD8 T cells it is expected that the response to SV40 T antigen would be tested. Should be the effect of FOXF2 on adenocarcinoma development, it might be possible of testing whether FOXF2 enlarge the time window in which SV40 is no yet tolerized. This might suggest a window of opportunity for active vaccination.

Finally, being the novelty of the study centered on fibroblast -produced FOXF2 immune effect it is not understandable the complete absence in Discussion of any immune arguments.

Other main points:

Increasing the size of RNA-scope and IHC pictures and adding arrows indicating positive cells would help comprehension.

Please specify which type(s) of statistical tests were used.

Figure 1D. The text claims that Lgr5 stromal cells specifically express Foxf2. However, RNA scope shows high number of Foxf2 cells not expressing Lgr5. Please reconcile.

Fig4. The authors say that FoxF2 inhibits the expression of Cxcl5, thus resulting in reduced accumulation of MDSC in tumor microenvironment (Table S1 and Figure 4A-E). Therefore, the rationale of silencing Cxcl5 in FoxF2 overexpressing cells (Figure 4F) to test “whether increasing stromal Foxf2 expression can still suppress tumor growth when Cxcl5 is low or absent” has little sense. Their demonstrated silencing of Cxcl5 in FoxF2 negative stromal cells already made the point. This part could be rewritten for clearness.

A Discussion on the possible ways active to increase FoxF2 in a therapeutic setting should be expanded.

Minor points:

Page 7, in commenting Figure 2F: PMN-MDSC are incorrectly indicated as Ly6C+

Page 7, commenting Figure S2K: histograms show myeloid cell frequency, nor their activity

Pies in figure 3F show increased frequency of metastasis in TRAMP mice overexpressing FoxF2, which is in contrast with what indicated in the text. Please reconcile

Reviewer #2 (Remarks to the Author): with expertise in Foxf2 biology, cancer

In this manuscript, the authors identified that the stromal specific expression of Foxf2 suppressed the proliferative capacity and lung metastases of prostate cancers. Mechanistically, Foxf2 repressed stroma-derived Cxcl5 secretion to decrease MDSC activity and enhance T cell immune function. As such, Foxf2 overexpression in stromal rendered prostate cancer sensitive to the immune checkpoint inhibitors. This study revealed a novel mechanism of immunosuppressive regulation in prostate cancer. There are some important issues and confusion should to be addressed.

Major comments:

1. For Fig. 1F, the authors point out that “Unfortunately, the antibody recognized the apical membrane of cancer cells in human prostate cancer specimens (white arrows, Fig. 1F and Supplementary Fig. 1B)”, and “reasoned that this staining was nonspecific”. It is strange that why it is nonspecific staining in cancer cells, but specific nuclear staining in stromal cells? The expression pattern and indicated molecular weight of FOXF2/foxf2 in prostate cancer cell lines with different biological characteristics and fibroblasts isolated from TZ and PZ of novel and cancer tissues, as well as in the cells with FOXF2/foxf2 knockdown or overexpression should be shown. The antibodies for FoXf2/FOXF2 protein staining or blotting in each figure should be written clearly. The molecular size of FoXf2/FOXF2 protein needs to be marked in WB images. The only marked molecular weight of FoXf2 in supplementary Fig. 3B is not correct for real FoXf2 protein.

How to determine the “relative fluorescence intensity of FOXF2 in prostate cancers of different Gleason score” in the dot plot? If only analyzed stromal cells, it should be described clearly. How about FOXF2 staining in cancer cells in different Gleason score? Images of SMA staining are showed. Since SMA is a typical CAF marker, how about the relationship between FOXF2 and SMA expression in stromal cells? Which one commercial antibody against FOXF2 was used in the listed three against FOXF2 antibodies in Supplementary Table 7 should be described. Immunofluorescence images derived from each individual grade also should be showed.

2. In Fig. 2, why the author only investigated whether increasing stromal Foxf2 expression affects prostate cancer cell growth in vitro and in vivo, but did not analyze the effects on invasive phenotype and metastatic ability of prostate cancer cells? Based on the data that “Coculturing the tumor cells with the Foxf2-overexpressing stromal cells did not affect the proliferation or apoptosis of the tumor cells in vitro (Data not shown)”, the authors cannot obtain the conclusion that “These studies exclude the possibility that stromal Foxf2 expression affects tumor cell biology via a paracrine stromal-epithelial interaction (line 134-137). The authors already summarized the role of Foxf2 in stromal-epithelial interaction via paracrine that is involved in cancer progression (line 73-75).

3. Since FOXF2 is higher expressed in normal fibroblasts and fibroblasts in benign tumor or low grad cancer tissues but lower expressed in CAFs in high grad cancer tissues (Fig. 1E and 1F), the experiments of loss-of-function in fibroblasts with high FOXF2 expression and gain-of-function in CAFs with low FOXF2 expression should be performed for Fig 2, and the effects on aggressive phenotype of cancer cells including migration and invasion in vitro as well as metastasis in vivo should be investigated in these experimental system. If the animal model of subcutaneous inoculation cannot develop into distant metastasis, the aggressive phenotype (e.g. angiogenesis, EMT, CSC) in the tumors should be examined.

4. In Fig. 3, the authors conclude that “the delayed tumor progression was a collective result of reduced proliferation and increased apoptosis in AP and DLP” (line 218-220), which is inconsistent with the results of Fig. 2. How to interpret this discrepancy?

5. In Fig. 4, same important issue as Fig. 2, the experiments of loss-of-function in fibroblasts with high FOXF2 expression and gain-of-function in CAFs with low FOXF2 expression also should be performed to validate the role of FOXF2/Foxf2 in regulating CXCLs/Cxcls. Importantly, how Foxf2 repressed stroma-derived Cxcl5 secretion to impair MDSCs recruitment but enhance T cell immune function? The receptors of Cxcl5 in immune cells should be detected.

6. In Fig. 6, the experiments of loss-of-function in fibroblasts should be performed to reveal whether leading to resistant to immune checkpoint inhibitors and promote tumor growth and metastasis.

7. In Fig. 7, RM-1 cells were injected through tail veins to investigate whether increased stromal Foxf2 in the lung can affect lung colonization. However, only tumor size is different in the lung, but not the rate of lung colonization (Fig. 7B and 7C). This result similar to the Fig. 2 that stromal Foxf2 suppresses tumor growth but not metastasis.

Other points:

1. There is wrong concept for β -Catenin in manuscript writing (line 51). It is a transcription cofactor but not a transcription factor.

2. There is inconsistent writing format for FOXF2 and Foxf2 in Fig. 1F and Fig. S1.

3. There is no “Fig. 1G” in Fig. 1, which is mentioned in line 118-119.

4. What is mean of Fig. 1D? It does not contribute the main story of this study.

5. It would be more informative if the analyses of MDSCs related markers and T cell function related markers were conducted by flow cytometry (Fig. 2H, Fig. 2I, Fig. 3J and Fig. 3K).

6. In Fig. 3F, “Met” and “No Met” may be marked inversely.

7. “Fig. 3O” mentioned in line 242 is a mistake label.

8. In Fig. 3L, myCAF only increased 6.7% in Col1a2-Foxf2-TRAMP mice (57.8% vs. 50.1%), however, why the authors give the results as “The percentage of myCAF increased by 15% in the Col1a2-Foxf2-TRAMP mice” (line 243)?

9. The gene names listed in Supplementary Table 4 should be standardized, e.g. “TGF β 1” should be TGFB1. The gene names of mouse and human need to be written correctly.

10. Abbreviations must be defined when they first appear, e.g. “TIME” (line 181) and “TRAMP” (line 182).

11. The molecular size of all proteins needs to be marked in WB images.

Reviewer #4 (Remarks to the Author): with expertise in scRNAseq, cancer immunology

In the manuscript “Stromal FOXF2 suppresses prostate cancer progression and metastasis by enhancing antitumor immunity” from Deyong Jia and collaborators they offer an extensive analysis using both mouse models and analysis of human samples, indicating that FOXF2 expression in stromal cells can positively influence the anti tumor immune response.

Most of my comments revolve around the statistical analysis, and method description of the computational analysis performed.

- 1) In lines 927-931: "Briefly, the immune and stromal datasets were handled separately with cells filtered by number of genes and percent mitochondrial genes, scaled and clustered, subjected to UMAP dimension reduction, and filtered to remove contaminating cell types based on cell type specific gene expression (e.g. epithelial cells). Cell type- or subtype-specific marker genes were used to identify cell types represented by clusters/subclusters and cell type annotations were appended to the metadata as was sample ID". Which markers were used for cell annotation? Are these the ones in Supplementary 3E?
- 2) Which genes were used for genes used for Fig. 3L cluster annotation? Are these the ones in Supplementary 3I? These are of high interest to the community and maybe add a note about their location in main figure legend?
- 3) Supplementary 3E: the UMAPs displayed with the gene expression of T cells are different from the upper most UMAPS, please check.
- 4) 3G lacks statistical analysis. 3H it is not clear how the analysis were performed. For 3M it reads in the methods "CAF signature scores were compared for control versus Foxf2 samples using the Wilcoxon rank sum test." How exactly was this calculated? Were the cluster divided between sample ID and the FindMarkers function used? Were each individual cell used as an individual sample? Or each animal? Can you please expand considerably in all the statistical analysis and code used for those. The same holds true for Supplementary 3K.

Reviewer #5 (Remarks to the Author): with expertise in stroma, prostate cancer

The authors have focused on the interesting role of Foxf2 in PCa. They showed that it is a stroma-specific factor with conserved role in human and mouse prostate. In clinical setting, Foxf2 expression is inversely linked to Gleason Score/ disease progression. Overexpression of Foxf2 in mouse prostate stromal cells reduced in vivo tumor growth of two PCa cell lines, but only in immunocompetent and genetic background, thus indicative of an immune cell modulatory role. To investigate the role of Foxf2 in in vivo tumorigenesis, a stromal cell specific-Foxf2 overexpressing mouse model was crossed with a prostate adenocarcinoma mouse model and extensively characterised in terms of tumor growth and tumor microenvironment (immune and stromal components). Interestingly, Foxf2 overexpression seemed to reduce tumor formation, enhance tumor cell infiltration and CAF composition. The immunomodulatory role of Foxf2 was found to be mediated by chemokine Cxcl5, and a beneficial combinatorial effect of Foxf2 overexpression and immunotherapy has been identified.

Major points:

- The introduction could use additional information on Foxf2 in prostate and rationale for the study.
- In Fig.1F, the entire panel has to be improved in terms of staining and resolution. If the antibody gives background in the epithelial glands, then it should be replaced with another one. Also, α SMA staining has nuclear pattern instead of cytoplasmic, so it should be optimised as well. Please show higher magnification images and merge each of the individual stainings with the DAPI to demonstrate localisation. In the X axis of the graph, is it correct "Gleason score" or is it groups 3 to 5? Please show a representative image for all conditions.
- To complement the correlation plot in Sup.Fig.1 I would recommend a prior quantification of stroma index on those tissues in parallel with the Foxf2 quantification.
- In Sup.Fig.3D, the pattern of SV40 expression does not overlap with the luminal AR+ cells, given that the TRAMP adenocarcinoma is initiated in probasin-positive luminal cells please provide evidence on which cell type is responsible for the phenotype (e.g. co-labeling SV40 with CK8/CK5/P63).
- the SV40 pattern is over all reduced in the Cola1a2-Foxf2-TRAMP model and particularly in the VP, how do the authors explain this, also with regards to the lack of size reduction in ventral lobe (Sup.Fig.3C) while the VP is the site of tumorigenic transformation in the TRAMP model?

- Regarding the presence of metastases in the Cola1a2-TRAMP and Cola1a2-Foxf2-TRAMP the authors state that the number of lung metastases is significantly reduced in the Cola1a2-Foxf2-TRAMP. However, in the Fig.3F pie charts, the occurrence of lung metastases in the Cola1a2-Foxf2-TRAMP is higher (12/12) than the Cola1a2-TRAMP model (10/17). Please correct this.
 - Please provide information on the selected markers that we were used to assign the CAF subclusters (iCAF, myCAFs and perivascular), as well as the CAF score, of the scRNA SEq data analysis of Fig.3 and Sup.Fig.3. How do the authors explain the increase of myofibroblastic CAF events detected, versus the overall reduction of CAF related gene expression in the Foxf2 overexpressing group? Please provide information in the discussion.
 - The analysis of the tumor models with and without Foxf2 overexpression, as well as those treated with immune checkpoint inhibitors, should be complemented with tissue stainings for CD3, CD8, PD1, PDL1, Foxf2, in order to demonstrate the in situ immune cell infiltration status and the co-localisation patterns with regards to tumor and stromal areas.
 - The enhancement of efficacy of anti-CTLA4 and anti-PD1 inhibitors is very interesting, however it is difficult to foresee a translational application for Foxf2 or Cxcl5 modulation. Data on the expression status of Fox2 and Cxcl5 from cohort of patients receiving immunotherapy should be included. Please include in your discussion this aspect regarding potential clinical applicability or how do these findings may be used for patient stratification for immunotherapy.
-
- Minor comments: In Fig.2 and Sup.Fig.2 please label the panels with the tumor type evaluated.
 - In the text line 242 the panel Fig.3O is referenced by not corresponding to the figure

We thank the reviewers for the constructive comments. We have addressed all the reviewers' concerns point-by-point by performing the requested experiments, correcting mistakes, clarifying confusions, and acknowledging the limitations. All changes were marked in red in the revised manuscript.

Reviewer #1

Major concerns

The statement: “On the C57BL/6 background, it (TRAMP) develops prostatic intraepithelial neoplasia (PIN) at 3 months and prostate cancer with focal neuroendocrine differentiation (particularly in ventral prostate) at 6 months” is to my knowledge incorrect. Differently from the FVB background, C57BL/6 TRAMP mice spontaneous primary neuroendocrine tumor occurs in less than 10% of mice in the ventral lobe.

Reply: the TRAMP mice in our lab develop focal neuroendocrine tumor in ventral prostate in approximately 46% of the mice at the age of 9 months. We apologize for leaving out this detail in the original submission. This frequency is within the range (19-67% in 6-mon-old mice) reported in two previous studies (PMIDs: 21360561, 18156212). C57BL/6 colonies from different vendors or housed in different institutes may affect this frequency. We have included that “In our C57BL/6 colonies, approximately 46% of TRAMP mice developed neuroendocrine prostate cancer (NEPC) at the 9 months of age. NEPCs in ventral prostate are likely developed *de novo* and tends to form focal and huge tumors (Supplementary Fig. 5a), whereas NEPCs in anterior and dorsolateral lobes are relatively rare and sporadic.” In the first paragraph on page 10. We thank the reviewer for pointing out this.

Does FOXF2 protect from NE primary transformation and/or NE trans-differentiation of existing adenocarcinoma or it has a distinct effect on the adenocarcinoma?

Reply: In our TRAMP colony, NEPCs in ventral prostate are likely developed *de novo* and tends to form focal and huge tumors, whereas NEPCs in anterior and dorsolateral lobes are relatively rare and sporadic. we revisited our original data by performing IHC analysis against synaptophysin and detected no difference in the NEPC frequency in the control and Foxf2-expressing group. NEPC is rare in anterior and dorsolateral lobes in our C57Bl/6 colonies. We also examined the expression of NEPC associated genes (*Syp*, *Chga*, and *Eno2*) in the adenocarcinoma tissues from TRAMP mice and did not observe significant changes in their expression. These results indicate that stromal Foxf2 expression does not affect NE primary formation or trans-differentiation in this model. These new data have been included as the Supplementary Figs. 5a-c and described in the first paragraph on page 10 in the revised manuscript.

Another feature of TRAMP mice is the notable expression of SV40, which is an effective endogenous tumor antigen, recognized by CD8 T lymphocytes and tolerized along tumor progression (Eur J Immunol. 35:66-75; 2005). In a paper aiming at demonstrating that the effect of FOXF2 pass through CD8 T cells it is expected that the response to SV40 T antigen would be tested. Should be the effect of FOXF2 on adenocarcinoma development, it might be possible of

testing whether FOXF2 enlarge the time window in which SV40 is no yet tolerized. This might suggest a window of opportunity for active vaccination.

Reply: We thank the reviewer for this interesting and constructive comment. In our experiments, we treated 8-week-old TRAMP;Col1a2-CreER;R26-LSL-Foxf2 mice with tamoxifen for 4 days to activate stromal Foxf2 expression. Therefore, by the time Foxf2 was expressed in the prostate stromal cells, the experimental mice were already about 9-10 weeks of age. At this time point, peripheral tolerance towards SV40 Tag-IV has already been almost fully developed based on the published study (Eur J Immunol. 35:66-75; 2005).

Nevertheless, we performed an additional experiment to address the reviewer's comment. We treated TRAMP and TRAMP;Col1a2-CreER;R26-LSL-Foxf2 mice with tamoxifen at 8 weeks of age to activate stromal Foxf2 expression. Then we immunized the mice at 11 weeks of age (young) and 6 months of age (old) with SV40 Tag-IV-pulsed dendritic cells. Splenocytes were isolated from these mice a week later and stimulated with SV40 Tag-IV *in vitro*. Our result indicated that the CD8⁺ T cells from the two groups responded similarly as determined by IFN γ production. This result indicates that stromal Foxf2 expression does not impact the tolerance of TRAMP model toward Tag-IV. These data have been included as the Supplementary Fig. 6 and described in the second paragraph on page 10 in the revised manuscript.

We want to thank this reviewer for this very constructive comment, which leads us learn more about immunology and makes our study more comprehensive.

Finally, being the novelty of the study centered on fibroblast -produced FOXF2 immune effect it is not understandable the complete absence in Discussion of any immune arguments.

Reply: We have included the discussion relating to immune tolerance to SV40 T and the potential therapeutic application of stromal Foxf2-mediated signaling in immune therapy in the last paragraph on page 17.

Minor concerns

Increasing the size of RNA-scope and IHC pictures and adding arrows indicating positive cells would help comprehension.

Reply: We have increased the size of the pictures and add additional representative images for cancer at different Gleason patterns in Figs. 1a, 1f and Supplementary Fig. 1b,1c. Arrows were added in Fig. 1a, 1f and Supplementary Fig. 1c.

Please specify which type(s) of statistical tests were used.

Reply: We have included statistical test in materials and methods and figure legends throughout the manuscript.

Figure 1D. The text claims that Lgr5 stromal cells specifically express Foxf2. However, RNA scope shows high number of FoxF2 cells not expressing Lgr5. Please reconcile.

Reply: Lgr5 is also highly expressed by the stromal cells in the proximal region based on our previous publications. It was used to locate proximal prostate in the RNAScope analysis. But we agree with the reviewer that our description was inaccurate and have modified the text in the first paragraph on page 5 as “An RNA-In-Situ duplex analysis of *Lgr5* and *Foxf2* further confirmed

that *Foxf2* is specifically expressed by the stromal cells and the expression level is higher in the mouse proximal prostatic ducts than in distal ducts”.

Fig4. The authors say that FoxF2 inhibits the expression of Cxcl5, thus resulting in reduced accumulation of MDSC in tumor microenvironment (Table S1 and Figure 4A-E). Therefore, the rationale of silencing Cxcl5 in FoxF2 overexpressing cells (Figure 4F) to test “whether increasing stromal Foxf2 expression can still suppress tumor growth when Cxcl5 is low or absent” has little sense. Their demonstrated silencing of Cxcl5 in FoxF2 negative stromal cells already made the point. This part could be rewritten for clearness.

Reply: We agree with the reviewer that silencing Cxcl5 in control stromal cells demonstrated its critical role. By silencing Cxcl5 in Foxf2-expressing stromal cells, we meant to test whether Cxcl5 is a major player through which increased stromal Foxf2 expression suppresses tumor growth. We apologize for the confusing text in the original submission and have revised the manuscript as “To determine whether Cxcl5 is a major player downstream of Foxf2, we investigated whether suppressing Cxcl5 can attenuate or ablate stromal Foxf2-mediated tumor suppressive effect.” in the fourth paragraph on page 12.

A Discussion on the possible ways active to increase FoxF2 in a therapeutic setting should be expanded.

Reply: we have included relating discussion in the last paragraph on page 17.

Page 7, in commenting Figure 2F: PMN-MDSC are incorrectly indicated as Ly6C+

Reply: we have corrected the mistake. Thanks for pointing this out.

Page 7, commenting Figure S2K: histograms show myeloid cell frequency, nor their activity

Reply: we have made the change according to the reviewer’s recommendation on page 8.

Pies in figure 3F show increased frequency of metastasis in TRAMP mice overexpressing FoxF2, which is in contrast with what indicated in the text. Please reconcile

Reply: we have corrected the mistake in Figure 3f. Thanks for pointing this out.

Reviewer #2

1. For Fig. 1F, the authors point out that “Unfortunately, the antibody recognized the apical membrane of cancer cells in human prostate cancer specimens (white arrows, Fig. 1F and Supplementary Fig. 1B)”, and “reasoned that this staining was nonspecific”. It is strange that why it is nonspecific staining in cancer cells, but specific nuclear staining in stromal cells? The expression pattern and indicated molecular weight of FOXF2/foxf2 in prostate cancer cell lines with different biological characteristics and fibroblasts isolated from TZ and PZ of novel and cancer tissues, as well as in the cells with FOXF2/foxf2 knockdown or overexpression should be shown. The antibodies for FoXf2/FOXF2 protein staining or blotting in each figure should be written clearly. The molecular size of FoXf2/FOXF2 protein needs to be marked in WB images. The only marked molecular weight of FoXf2 in supplementary Fig. 3B is not correct for real FoXf2 protein.

Reply: we tested three commercially available Foxf2 antibodies (AF6988 from R&D; MBS2523449 from MyBioSource; ab198283 from Abcam) and a rabbit antibody against human FOXF2 that we generated through Sino Biological Inc. The one we used in this manuscript (AF6988 from R&D) is the one that works the best for immunostaining, even though it stains nonspecifically in cancer cells. We are fully aware of the limitation of this antibody. Therefore, in our original submission we performed multiple control experiments. First, we showed using the wild type and Foxf2 null embryos and demonstrated that the antibody displayed a nuclear staining in the WT samples but did not stain in the Foxf2 null samples. Second, we used RNAscope assay to confirm that the *FOXF2* transcript is present only in prostate stromal cells but not epithelial cells. This is to support that the apical and cytoplasmic immunostaining of FOXF2 in epithelial cells is not specific. Finally, FOXF2 staining in the stromal cells display an expected nuclear pattern. In addition, we specifically focus on the prostate stromal cells and showed that the *FOXF2* transcript level correlates with protein level in these cells. Based on these studies, we concluded that the staining in the stromal cells is likely Foxf2 specific. It is unfortunate that a perfect antibody for Foxf2 is not available. Similar problem has been encountered by other Foxf2 researchers. For example, in a study (PMCID: 30561059) at Dev Cell by Peter Carlsson’s group, the authors stated clearly on page 21 that the antibody has a strong cross-reactivity with a pan-neuronal epitope. We hope the reviewer appreciate our effort and agree that we have done all we could do using the best available reagent.

To further answer the reviewer’s question, we have shown that the two Foxf2 antibodies (AF6988 and MBS2523449) that work for Western Blot analysis and an antibody against the FLAG tag produced signaling of the same size in prostate stromal cells and RM-1 cells overexpressing a FLAG-tagged Foxf2 (Supplementary Fig. 1a). The apparent molecular weight is approximately 55kD, which is the same as we showed in the previous Supplementary Fig. 3b (now Supplementary Fig. 4b). This size is also consistent with that shown in several previously published studies (PMCID: 4712829, 31222004, 30561059). We have included the size of the Foxf2 protein in Western blot assays in Fig. 5f and Supplementary Figs. 1a, 4b, as the reviewer has requested.

To further support our finding, we performed an additional analysis using laser capture microscopy. We captured stromal cells from prostate cancer specimens of different Gleason patterns and confirmed that Foxf2 expression inversely correlates with disease grades. This new data is described the first paragraph on page 6 and included in the revised manuscript as Fig.1g.

How to determine the “relative fluorescence intensity of FOXF2 in prostate cancers of different Gleason score” in the dot plot? If only analyzed stromal cells, it should be described clearly. How about FOXF2 staining in cancer cells in different Gleason score? Images of SMA staining are showed. Since SMA is a typical CAF marker, how about the relationship between FOXF2 and SMA expression in stromal cells? Which one commercial antibody against FOXF2 was used in the listed three against FOXF2 antibodies in Supplementary Table 7 should be described. Immunofluorescence images derived from each individual grade also should be showed.

Reply: We have now indicated clearly on page 6 and in the figure legend that the relative fluorescent intensity of FOXF2 in prostate cancers of different Gleason patterns in Supplementary Fig. 1d was specifically determined in the stromal cells. How the signaling is quantified was described in detail in the method section on page 44.

FOXF2 staining in cancer cells is not nuclear staining but rather apical or cytoplasmic. We previously shown by in Situ that FOXF2 is mainly expressed in the stromal cells in human prostate specimens (Supplementary Fig. 1c). During the revision, we further performed laser capture microscopic analysis and confirmed that *FOXF2* is mainly expressed in the stromal compartment (Supplementary Fig. 1e).

Although SMA is a marker for CAF, it alone is not sufficient to faithfully represent stromal cell status. That is why a collection of markers are used to define stromal cell status in published studies. To address the reviewer’s curiosity, we examined the expression of *SMA* in prostate cancer specimens of different Gleason patterns, our data shown here indicate that *SMA* expression is decreased in the stroma of advanced prostate cancer and there seems to be a positive correlation between *FOXF2* and *SMA*, although FOXF2 knockdown was reported to transcriptionally upregulates *SMA* (PMCID: 25631042). This data is included in the revised manuscript as Supplementary Fig. 10.

Representative immunostaining images for individual cancer grade is included now in Fig. 1f. The information of FOXF2 antibody is added in the legend of Fig. 1f and Supplementary Table 8.

2. In Fig. 2, why the author only investigated whether increasing stromal Foxf2 expression affects prostate cancer cell growth in vitro and in vivo, but did not analyze the effects on invasive phenotype and metastatic ability of prostate cancer cells? Based on the data that “Coculturing the tumor cells with the Foxf2-overexpressing stromal cells did not affect the proliferation or apoptosis of the tumor cells in vitro (Data not shown)”, the authors cannot obtain the conclusion that “These studies exclude the possibility that stromal Foxf2 expression affects tumor cell biology via a paracrine stromal-epithelial interaction (line 134-137). The authors already summarized the role of Foxf2 in stromal-epithelial interaction via paracrine that is involved in cancer progression (line 73-75).

Reply: we apologize for the inaccurate conclusion. Our study reveals a novel role of stromal Foxf2 in regulating tumor immunity but is not meant to challenge a role of paracrine signaling mediated by stromal Fox2 via stromal-epithelial interaction in tumor progression. We have revised the conclusion on page 6 as “These studies showed that stromal Foxf2 expression did

not significantly affect the growth and survival of prostate cancer cells via a paracrine stromal-epithelial interaction in these models".

3. Since FOXF2 is higher expressed in normal fibroblasts and fibroblasts in benign tumor or low grad cancer tissues but lower expressed in CAFs in high grad cancer tissues (Fig. 1E and 1F), the experiments of loss-of-function in fibroblasts with high FOXF2 expression and gain-of-function in CAFs with low FOXF2 expression should be performed for Fig 2, and the effects on aggressive phenotype of cancer cells including migration and invasion *in vitro* as well as metastasis *in vivo* should be investigated in these experimental system. If the animal model of subcutaneous inoculation cannot develop into distant metastasis, the aggressive phenotype (e.g. angiogenesis, EMT, CSC) in the tumors should be examined.

Reply: we performed additional experiment per the reviewer's comment and demonstrated that knocking down stromal Foxf2 led to an increase in tumor size. Decreasing Foxf2 in stromal cells also inhibited MDSC and increased T cell infiltration as predicted. The new data has been included as Supplementary Figs. 3d, 3e, and described in the last paragraph on page 7 in the revised manuscript.

Per the reviewer's request, we performed *in vitro* migration and invasion assay using Matrigel chamber (Supplementary Fig.2d), measured vessel density (Supplementary Fig.2g) in tumor xenograft by immunostaining of CD31, and measured representative markers for putative cancer stem cells and EMT (Supplementary Figs. 2h, 2i). These results indicate that stromal Foxf2 does not alter tumor growth and survival via these signaling in the models that we tested.

As mentioned in our response to question 2, based on the results from immunodeficient and immunocompetent models, our study demonstrated a definitive role of stromal Foxf2 in regulating tumor immunity. Our study is not meant to challenge the role of paracrine signaling mediated by stromal Fox2 via stromal-epithelial interaction in tumor progression. We also included discussion regarding this in the last paragraph on page 17.

4. In Fig. 3, the authors conclude that "the delayed tumor progression was a collective result of reduced proliferation and increased apoptosis in AP and DLP" (line 218-220), which is inconsistent with the results of Fig. 2. How to interpret this discrepancy?

Reply: The tumor cells in more advanced tumor proliferated faster. The proliferating index in older TRAMP mice is higher than that in the younger TRAMP mice. Therefore, in the TRAMP;Col1a2-CreER;R26-LSL-Foxf2 model, the reduced proliferation likely reflects the delayed disease progression during the 10-month period. In contrast, in the fast developed (within 2 weeks) syngeneic aggressive RM-1 and Pten-Kras models, no such difference can be detected. We have changed our conclusion on page 9 to make it more accurate: "Consistent with the delayed tumor progression, there was a reduced proliferating index of tumor cells determined by immunostaining of BrdU in AP and DLP (Fig. 3d). Or alternatively, stromal Foxf2 may mediate additional signaling that suppress tumor cell proliferation in the TRAMP model."

5. In Fig. 4, same important issue as Fig. 2, the experiments of loss-of-function in fibroblasts with high FOXF2 expression and gain-of-function in CAFs with low FOXF2 expression also should be performed to validate the role of FOXF2/Foxf2 in regulating CXCLs/Cxcls. Importantly, how Foxf2 repressed stroma-derived Cxcl5 secretion to impair MDSCs recruitment

but enhance T cell immune function? The receptors of Cxcl5 in immune cells should be detected.
Reply: We showed that knocking down Foxf2 in mouse prostate stromal cells upregulates Cxcl5. The new data has been included as Supplementary Fig. 8a and described on page 12 in the revised manuscript.

Regulation of MDSC by Cxcl5 and regulation of T cell immune function by MDSC cells have been reported and widely accepted in the field (PMID: 19197294). We examined the immune cells by qRT-PCR and showed that Cxcr2 is mostly expressed by the immune cells in the tumor tissues (Supplementary Fig. 8b, page 12).

6. In Fig. 6, the experiments of loss-of-function in fibroblasts should be performed to reveal whether leading to resistant to immune checkpoint inhibitors and promote tumor growth and metastasis.

Reply: Based on the studies from ours as well as previous publications (PMCID: 23983257, 26787820), the RM-1 model is irresponsive to anti PD-1 and has limited response to anti-CTLA4. Therefore, it is not feasible to prove that knocking down Foxf2 further enhances irresponsiveness of the model to the ICB therapies.

7. In Fig. 7, RM-1 cells were injected through tail veins to investigate whether increased stromal Foxf2 in the lung can affect lung colonization. However, only tumor size is different in the lung, but not the rate of lung colonization (Fig. 7B and 7C). This result similar to the Fig. 2 that stromal Foxf2 suppresses tumor growth but not metastasis.

Reply: We also determined the colony number and the result showed that the number of the colonies was also decreased. This additional result is described in the text and included in the revised Fig. 7c.

Other points:

1. There is wrong concept for β -Catenin in manuscript writing (line 51). It is a transcription cofactor but not a transcription factor.

Reply: we have corrected this.

2. There is inconsistent writing format for FOXF2 and Foxf2 in Fig. 1F and Fig. S1.

Reply: We use FOXF2 and Foxf2 for human and mouse genes, respectively. We apologize for the confusion.

3. There is no "Fig. 1G" in Fig. 1, which is mentioned in line 118-119.

Reply: This has been corrected.

4. What is mean of Fig. 1D? It does not contribute the main story of this study.

Reply: This information is to highlight the similarity between the human and mouse prostate. The information may not be useful for general audience but is crucial for the researchers in the prostate cancer field. We hope the reviewer understand. Thanks.

5. It would be more informative if the analyses of MDSCs related markers and T cell function related markers were conducted by flow cytometry (Fig. 2H, Fig. 2I, Fig. 3J and Fig. 3K).

Reply: We did use FACS to measure IFN γ and GZMB to verify that T cell activity was

increased. We agree with the reviewer that analyzing MDSC parameters using FACS can provide additional information, but also feel that our current result is also from standard analysis used in the field that can sufficiently support our conclusion (PMID: 28321130). Therefore, we did not repeat these experiments due to limited resource and personnel. We hope the reviewer can understand.

6. In Fig. 3F, “Met” and “No Met” may be marked inversely.

Reply: Thanks for catching this mistake and we have made the correction.

7. “Fig. 3O” mentioned in line 242 is a mistake label.

Reply: This has been updated.

8. In Fig. 3L, myCAF only increased 6.7% in *Col1a2-Foxf2-TRAMP* mice (57.8% vs. 50.1%), however, why the authors give the results as “The percentage of myCAF increased by 15% in the *Col1a2-Foxf2-TRAMP* mice” (line 243)?

Reply: The percentage increased from 50.1% to 57.8%, which is increased by 15%. To make the description more instinctive, we rephrased as “The percentage of myCAF (57.8%) in the *Col1a2-Foxf2-TRAMP* mice is higher than that of the control mice (50.1%)” on page 11.

9. The gene names listed in Supplementary Table 4 should be standardized, e.g. “TGFβ1” should be TGFBI. The gene names of mouse and human need to be written correctly.

Reply: Thank you for pointing this out. We have updated human and mouse gene names accordingly.

10. Abbreviations must be defined when they first appear, e.g. “TIME” (line 181) and “TRAMP” (line 182).

Reply: TIME was defined in the first paragraph in Introduction. Full name for TRAMP is mentioned on page 8.

11. The molecular size of all proteins needs to be marked in WB images.

Reply: we have updated this information in all Figures in the revised manuscript.

Reviewer #4

1) In lines 927-931: “Briefly, the immune and stromal datasets were handled separately with cells filtered by number of genes and percent mitochondrial genes, scaled and clustered, subjected to UMAP dimension reduction, and filtered to remove contaminating cell types based on cell type specific gene expression (e.g. epithelial cells). Cell type- or subtype-specific marker genes were used to identify cell types represented by clusters/subclusters and cell type annotations were appended to the metadata as was sample ID”. Which markers were used for cell annotation? Are these the ones in Supplementary 3E?

Reply: We thank this reviewer for drawing our attention to the methods on cluster annotation. We have added text to the methods section stating which genes were primarily used for cluster and subcluster annotation. We have clarified in the Fig. 3 legend and in the methods that the genes presented in Supplementary Fig. 7a (previous Fig. S3E) were used to annotate the clusters in Fig. 3g.

2) Which genes were used for genes used for Fig. 3L cluster annotation? Are these the ones in Supplementary 3I? These are of high interest to the community and maybe add a note about their location in main figure legend?

Reply: We have clarified in the Fig. 3 legend and in the methods that the genes presented in Supplementary Fig. 7e (previous Fig. S3I) were used to annotate the clusters in Fig. 3l. We thank the reviewer for encouraging us to highlight these important data.

3) Supplementary 3E: the UMAPs displayed with the gene expression of T cells are different from the upper most UMAPS, please check.

Reply: We thank the reviewer for the careful examination of our manuscript. We have corrected the error in the plots identified by the reviewer.

4) 3G lacks statistical analysis. 3H it is not clear how the analysis were performed. For 3M it reads in the methods “CAF signature scores were compared for control versus Foxf2 samples using the Wilcoxon rank sum test.” How exactly was this calculated? Were the cluster divided between sample ID and the FindMarkers function used? Were each individual cell used as an individual sample? Or each animal? Can you please expand considerably in all the statistical analysis and code used for those. The same holds true for Supplementary 3K.

Reply: We have added to our discussion of statistics and analysis function parameters throughout the figure legends and methods to clarify how the analyses were performed. Specifically, we now state the FindAllMarkers function was implemented to detect differential gene expression using a Wilcoxon Rank Sum test for Figure 3h and Supplementary Figure 7g (previous Fig. S3k). For Figure 3m we have clarified that CAF signature scores for the two populations were also compared using a Wilcoxon Rank Sum test and taking individual cells as samples for populations isolated from Foxf2 and control mice. We have added to our discussion of statistics and analysis function parameters throughout in the revised manuscript.

Reviewer #5

Major points:

- The introduction could use additional information on Foxf2 in prostate and rationale for the study.

Reply: there is limited previous studies regarding the role of stromal Foxf2 in prostate cancer. These studies were discussed in the last paragraph on page 3. We became interested in Foxf2 because we independently identified it as a protein highly expressed by the stromal cells in TZ and we want to test the hypothesis that prostate cancer rarely develops in TZ because of its unique stromal microenvironment. This rationale was described in the second and third paragraph of our introduction.

- In Fig.1F, the entire panel has to be improved in terms of staining and resolution. If the antibody gives background in the epithelial glands, then it should be replaced with another one. Also, α SMA staining has nuclear pattern instead of cytoplasmic, so it should be optimised as well. Please show higher magnification images and merge each of the individual stainings with the DAPI to demonstrate localisation. In the X axis of the graph, is it correct "Gleason score" or is it groups 3 to 5? Please show a representative image for all conditions.

Reply: we tested three commercially available Foxf2 antibodies (AF6988 from R&D; MBS2523449 from MyBioSource; ab198283 from Abcam) and a rabbit antibody against human FOXF2 that we generated through Sino Biological Inc. The one we used in this manuscript (AF6988 from R&D) is the one that works the best for immunostaining, even though it stains nonspecifically in cancer cells. We are fully aware of the limitation of this antibody. Therefore, in our original submission we performed multiple control experiments. First, we showed using the wild type and Foxf2 null embryos and demonstrated that the antibody displayed a nuclear staining in the WT samples but did not stain in the Foxf2 null samples. Second, we used RNAscope assay to confirm that the *FOXF2* transcript is present only in prostate stromal cells but not epithelial cells. This is to support that the apical and cytoplasmic immunostaining of FOXF2 in epithelial cells is not specific. Finally, FOXF2 staining in the stromal cells display an expected nuclear pattern. In addition, we specifically focus on the prostate stromal cells and showed that the *FOXF2* transcript level correlates with protein level in these cells. Based on these studies, we concluded that the staining in the stromal cells is likely Foxf2 specific. It is unfortunate that a perfect antibody for Foxf2 is not available. Similar problem has been encountered by other Foxf2 researchers. For example, in a study (PMCID: 30561059) at Dev Cell by Peter Carlsson's group, the authors stated clearly on page 21 that the antibody has a strong cross-reactivity with a pan-neuronal epitope. We hope the reviewer appreciate our effort and agree that we have done all we could do using the best available reagent.

To further support our finding, we performed an additional analysis using laser capture microscopy. We captured stromal cells from prostate cancer specimens of different grades and confirmed that Foxf2 expression inversely correlates with disease grades. This new data is included in the revised manuscript as new Fig. 1g.

We have optimized α SMA staining and provided higher magnification images with merged signaling as requested by the reviewer in Fig. 1f. We thank the reviewer for the comment

regarding “Gleason score”. We have changed it to “Gleason pattern” in Figs. 1f and 1g. Representative images for all Gleason patterns are included now in Fig. 1f.

- To complement the correlation plot in Sup.Fig.1 I would recommend a prior quantification of stroma index on those tissues in parallel with the Foxf2 quantification.

Reply: the correlation was indeed performed specifically in the prostate stromal cells. We apologize for not making this clearer in the original submission. We have stated on page 6 and in the legend of Supplemental Fig. 1 that the analysis was performed specifically in the stromal cells.

In addition, as mentioned above, we also performed an additional analysis using laser capture microscopy. We captured stromal cells from prostate cancer specimens of different patterns and confirmed that Foxf2 expression inversely correlates with Gleason patterns. This new data is included in the revised manuscript as the new Fig. 1g.

- In Sup.Fig.3D, the pattern of SV40 expression does not overlap with the luminal AR+ cells, given that the TRAMP adenocarcinoma is initiated in probasin-positive luminal cells please provide evidence on which cell type is responsible for the phenotype (e.g. co-labeling SV40 with CK8/CK5/P63).

Reply: We have included clearer and more representative images in Sup. Fig. 4d (previous Fig. S3d) showing that SV40 T antigen is expressed mainly in the AR-expressing CK8 positive luminal cells.

- the SV40 pattern is over all reduced in the Cola1a2-Foxf2-TRAMP model and particularly in the VP, how do the authors explain this, also with regards to the lack of size reduction in ventral lobe (Sup.Fig.3C) while the VP is the site of tumorigenic transformation in the TRAMP model?

Reply: We apologize for the quality of the images in the original submission. Expression of SV40 T was not altered in the prostate Col1a2-Foxf2-TRAMP model. We have included new images of SV40 T staining in different prostate lobes of the mice in the two group in Supplementary Fig. 4d.

We reported in the original submission that no significant difference was noted in term of tissue size and immune cell composition in VP. TRAMP mice develop tumor more efficiently in AP and DLP. VP tumor is relatively smaller though a fraction of mice developed large de novo NEPC nodules in VP. To exclude the size variance, we only analyzed VPs from those mice without de novo NEPC. We do not know why no change was noted in VP. We have included on page 11 that “It remains unclear why tumor growth and immune microenvironment in VP was not significantly affected.”.

- Regarding the presence of metastases in the Cola1a2-TRAMP and Cola1a2-Foxf2-TRAMP the authors state that the number of lung metastases is significantly reduced in the Cola1a2-Foxf2-TRAMP. However, in the Fig.3F pie charts, the occurrence of lung metastases in the Cola1a2-Foxf2-TRAMP is higher (12/12) than the Cola1a2- TRAMP model (10/17). Please correct this.
Reply: We apologize for the mistake and have corrected it.

- Please provide information on the selected markers that we were used to assign the CAF

subclusters (iCAF, myCAFs and perivascular), as well as the CAF score, of the scRNA SEq data analysis of Fig.3 and Sup.Fig.3. How do the authors explain the increase of myofibroblastic CAF events detected, versus the overall reduction of CAF related gene expression in the Foxf2 overexpressing group? Please provide information in the discussion.

Reply: We have clarified in the legend of Fig. 3 and in the methods that the genes presented in Supplementary Fig. 7a were used to annotate the clusters in Fig. 3g and the genes in Supplementary Fig. 7e were used to annotate the clusters in Fig. 3l. We've also listed the 30 genes used to general CAF signature scores in the legend of Fig. 3.

Annotation of myCAF and iCAF is based on the expression of previously published representative genes shown in Supplementary Fig. 7e. Foxf2 expression influences the population among the CAF population (myCAF versus iCAF). Nevertheless, the overall expression of the majority of the 30 CAF-related genes were downregulated, which leads to our conclusion that the CAF signature is downregulated. These two observations are not contradictory to each other.

- The analysis of the tumor models with and without Foxf2 overexpression, as well as those treated with immune checkpoint inhibitors, should be complemented with tissue stainings for CD3, CD8, PD1, PDL1, Foxf2, in order to demonstrate the in situ immune cell infiltration status and the co-localisation patterns with regards to tumor and stromal areas.

Reply: Because of the imperfect FOXF2 antibody and the cumbersome analysis of immunostaining, we decided to use laser capture microscopy in concert with qRT-PCR to address the reviewer's request. We captured cells in the stromal regions of prostate cancer specimens of different Gleason patterns and determine the expression of *FOXF2* and genes of interests and then investigated the correlation. We identified a significant positive correlation between *FOXF2* and *CD8a*, although there is no significant correlation between *FOXF2* and *CD3*. We also confirmed the significant inverse correlations between *FOXF2* and the M2 macrophage markers *CD163* and *CD206*, but not with *CD86*. There is also a trend for an inversed correlation between *FOXF2* and *S100A* although it did not reach statistical significance. Finally, *FOXF2* expression is also inversely correlated with those of *CXCL6*, *CXCL8*, and *PDCD1*. The new data is described on page 15 and included in the revised manuscript as Supplementary Fig. 10.

- The enhancement of efficacy of anti-CTLA4 and anti-PD1 inhibitors is very interesting, however it is difficult to foresee a translational application for Foxf2 or Cxcl5 modulation. Data on the expression status of Fox2 and Cxcl5 from cohort of patients receiving immunotherapy should be included. Please include in your discussion this aspect regarding potential clinical applicability or how do these findings may be used for patient stratification for immunotherapy.

Reply: There is only one publicly available dataset including 8 patients with metastatic prostate cancers treated with anti-PD-1 (5 non-responders and 3 responders) (PMID: 35322234). We contacted the authors to request access for data but never received a response. Meanwhile, we also doubt that the RNA-seq data from this small-scale study is helpful because the expression of *FOXF2* in the dataset was not from stromal cells in the prostate.

We have included discussion regarding potential clinical applicability or how do our findings may be used for patient stratification for immunotherapy on page 17.

Minor comments:

In Fig.2 and Sup.Fig.2 please label the panels with the tumor type evaluated.

Reply: We have included the labels in Fig. 2a, 2b, S2e, S2f, S2j, S2k.

In the text line 242 the panel Fig.3O is referenced by not corresponding to the figure

Reply: We have corrected the mistake.

REVIEWERS' COMMENTS

Reviewer #1 (Remarks to the Author):

I commended the authors for the additional work done to respond to this Reviewer. As a note of clearness i would suggest of not starting the Discussion claiming that FOXGF2 instigate antitumor immunity but, according the data presented, hampers the generation of an immunosuppressive microenvironment and therefore facilitate antitumor immunity.

Reviewer #2 (Remarks to the Author):

The revised version has been significantly improved according to the comments of reviewers. This study shows a novel role of stromal Foxf2 in regulating tumor immunity which helps to understand the effect of immune microenvironment on the occurrence and progression of prostate cancer.

Reviewer #4 (Remarks to the Author):

The authors have addressed my points during the revision.

Reviewer #5 (Remarks to the Author):

The manuscript has been overall improved. However, some aspects should be further elaborated. The translational relevance and main novelty of the paper must be more clearly presented. The decrease of Foxf2 gene expression in advanced stages shown and the proposed role in homeostasis vs tumor formation and immune modulation might be consequential and not causal event of just one factor. Are there genomic aberrations and epigenetic changes of Foxf2 in different prostate cancer patients and GEMMs? Which are the upstream regulators of Foxf2 expression and are they aberrant or mutated in PCa ? How do the authors explain and fit the role of Foxf2 in the existing landscape of immune suppression in prostate tumors.

Given that it is a key part of the study, the immune modulation and alteration of fibroblast subtypes by Foxf2 must be additionally proven and shown in situ in tumors by means of protein expression to complement the RNA expression from laser-capture microdissected stromal areas. Localisation of immune infiltration in the different anatomical areas of the prostate in the foxf2 overexpressing areas must be shown.

The study must be complemented by demonstrating the direct role of Foxf2 in tumorigenesis. for instance reverting the expression of Foxf2 by in vivo siRNA-mediated downregulation in the Col1a1-Foxf2 overexpressing tumors.

Please clearly indicate the list of signature genes, the studies based on which the genes of myCAF and other subclusters were used as well as rationale for their selection.

Response to reviewers' comments

Reviewer #1

As a note of clearness I would suggest of not starting the Discussion claiming that FOXGF2 instigate antitumor immunity but, according the data presented, hampers the generation of an immunosuppressive microenvironment and therefore facilitate antitumor immunity.

Reply: We thank the reviewer's constructive comment. We have made the change accordingly by stating: Our study shows that a higher stromal Foxf2 expression hampers the generation of an immunosuppressive microenvironment and facilitates antitumor immunity.

Reviewer #5

The translational relevance and main novelty of the paper must be more clearly presented. The decrease of Foxf2 gene expression in advanced stages shown and the proposed role in homeostasis vs tumor formation and immune modulation might be consequential and not causal event of just one factor. Are there genomic aberrations and epigenetic changes of Foxf2 in different prostate cancer patients and GEMMs? Which are the upstream regulators of Foxf2 expression and are they aberrant or mutated in PCa ? How do the authors explain and fit the role of Foxf2 in the existing landscape of immune suppression in prostate tumors.

Reply:

- (a) The translational relevance and novelty of our work were clearly stated in both abstract and discussion and were recognized by the other two reviewers.
- (b) In terms of the relationship between stromal Foxf2 expression and tumor grade, the reviewer raised a chicken and egg problem. Using human specimens, we cannot answer whether stromal Foxf2 expression is decreased because of tumor progression or tumor progression is faster because stromal Foxf2 expression is low. This is a question applicable to all tumor markers but does not affect their prognostic and even therapeutic application if their functional significance can be validated. In our case, our experiments using overexpressing and knocking down models clearly support that stromal Foxf2 expression can affect tumor growth by modulating tumor microenvironment. Therefore, we feel that the functional significance of stromal Foxf2 expression level is strongly supported by our data.
- (c) Genomic alterations in the stromal microenvironment of human prostate cancer are rare (PMID: 26753621). Therefore, it is likely that the expression of Foxf2 is regulated at the transcriptional level. Epithelial Shh has been shown to induce stromal Foxf2 expression. Basal cells are the main source of Shh in the prostate. Loss of basal cells occurs during prostate cancer initiation. This is one potential mechanism how stromal Foxf2 in the prostate is regulated. We have discussed this in the revised manuscript already.
- (d) It is not quite clear to us what the reviewer means when asking: “How do the authors explain and fit the role of Foxf2 in the existing landscape of immune suppression in prostate tumors.”

Given that it is a key part of the study, the immune modulation and alteration of fibroblast subtypes by Foxf2 must be additionally proven and shown in situ in tumors by means of protein expression to complement the RNA expression from laser-capture microdissected stromal areas. Localisation of immune infiltration in the different anatomical areas of the prostate in the foxf2 overexpressing areas must be shown.

Reply:

- (a) There is no single marker for individual fibroblast subtypes. All current studies rely on the expression of a group of genes by scRNA-seq analysis to define fibroblast subtypes. In the scRNA-seq analysis, we have a list of 30 CAF associated genes. We have not seen any published study in which only one or two antigens were used to define the changes of CAF phenotype quantitatively and definitively by IHC.
- (b) For immune cells, because of the limitation of the Foxf2 antibody that we described in the manuscript in Fig. 1, during revision, we have further used laser capture microscopy and demonstrated that the expression level of stromal *FOXF2* positively correlates with that of *CD8A* and inversely correlates with those of *CD206* and *CD163*. These markers are strictly immune cell markers; therefore, the expression levels should faithfully reflect the abundance of the

corresponding immune cells. We feel that this data is sufficient to demonstrate the correlation between stromal FOXF2 expression and major cell populations of immune microenvironment.

The study must be complemented by demonstrating the direct role of Foxf2 in tumorigenesis. for instance reverting the expression of Foxf2 by in vivo siRNA-mediated downregulation in the Col1a1-Foxf2 overexpressing tumors.

Reply: If we knock out a small LncRNA and notice some biological outcome, we will agree that it is reasonable to add back the sequence to confirm that the biology is indeed due to the LncRNA itself but not due to alteration of gene expression in adjacent genomic foci. In our case, we activated transgenic Foxf2 in the ROSA locus. We are not sure why the reviewer would want us to do this complement experiment, especially considering that it is not an easy task to use in vivo siRNA-mediated downregulation in the Col1a1-Foxf2-TRAMP tumors. In addition, we have performed both overexpression and knockdown experiments using the other two syngeneic tumor models and obtained the same conclusion. Therefore, we feel that our conclusion is sufficiently supported by the current functional studies.

Please clearly indicate the list of signature genes, the studies based on which the genes of myCAF and other subclusters were used as well as rationale for their selection.

Reply: The names of the genes were already listed in both Methods and main text and are mostly from the reference 44.